# Exploiting Curvature in Online Convex Optimization with Delayed Feedback

**Hao Qiu** [* 1]   **Emmanuel Esposito** [* 1]   **Mengxiao Zhang** [* 2]

## Abstract

In this work, we study the online convex optimization problem with curved losses and delayed feedback. When losses are strongly convex, existing approaches obtain regret bounds of order $d_{\max} \ln T$, where $d_{\max}$ is the maximum delay and $T$ is the time horizon. However, in many cases, this guarantee can be much worse than $\sqrt{d_{\text{tot}}}$ as obtained by a delayed version of online gradient descent, where $d_{\text{tot}}$ is the total delay. We bridge this gap by proposing a variant of follow-the-regularized-leader that obtains regret of order $\min\{\sigma_{\max} \ln T, \sqrt{d_{\text{tot}}}\}$, where $\sigma_{\max}$ is the maximum number of missing observations. We then consider exp-concave losses and extend the Online Newton Step algorithm to handle delays with an adaptive learning rate tuning, achieving regret $\min\{d_{\max} n \ln T, \sqrt{d_{\text{tot}}}\}$ where $n$ is the dimension. To our knowledge, this is the first algorithm to achieve such a regret bound for exp-concave losses. We further consider the problem of unconstrained online linear regression and achieve a similar guarantee by designing a variant of the Vovk-Azoury-Warmuth forecaster with a clipping trick. Finally, we implement our algorithms and conduct experiments under various types of delay and losses, showing an improved performance over existing methods.

## 1. Introduction

Online convex optimization (OCO) is a powerful framework for sequential decision making in uncertain environments (Hazan et al., 2007; Orabona, 2025). In classic OCO, a learner repeatedly makes a decision, incurs a loss for the chosen action, and uses the feedback of the loss function at

*Equal contribution  [1]Università degli Studi di Milano  [2]University of Iowa.  Correspondence to:  Hao Qiu <hao.qiu@unimi.it>, Emmanuel Esposito <emmanuel@emmanuelesposito.it>, Mengxiao Zhang <mengxiao-zhang@uiowa.edu>.

*Proceedings of the 42$^{nd}$ International Conference on Machine Learning*, Vancouver, Canada. PMLR 267, 2025. Copyright 2025 by the author(s).

this round to update her strategy in the next round. However, in many real-world applications, feedback is not immediately available after the learner's decision but is instead subject to a delay. For instance, in online ad recommendation systems (He et al., 2014), click-through information may be delayed, and during this time the system must continue making recommendations for other users without access to the delayed feedback.

Another crucial element in OCO is given by properties of the loss functions such as the curvature. It is indeed often the case that losses have additional curvature properties such as strong convexity or exp-concavity. For example, exp-concave losses are prevalent in portfolio management (Cover, 1991), in which the learner (investor) needs to distribute her wealth over a set of financial instruments in order to maximize her return. When the loss functions have a certain curvature, previous works (Hazan et al., 2007) have shown that a significantly better regret guarantee can be achieved (i.e., the so-called fast rates). However, this type of assumption received little attention when assuming that the feedback suffers some delay. Therefore, we are interested in investigating the following question:

*Can we design algorithms that exploit the loss curvature to obtain improved guarantees even with delayed feedback?*

There is a line of works studying OCO with delayed feedback. For general convex functions, Quanrud & Khashabi (2015) provided an algorithm called Delayed Online Gradient Descent (DOGD) and achieves a regret of $\mathcal{O}(\sqrt{T + d_{\text{tot}}})$ where $T$ is the time horizon and $d_{\text{tot}}$ is the total delay. Subsequently, Wan et al. (2022a); Wu et al. (2024) focused on strongly convex losses, introducing DOGD-SC and SDMD-RSC, which achieve a regret bound of $\mathcal{O}((d_{\max} + 1) \ln T)$, where $d_{\max}$ represents the maximum delay for any single round of feedback. However, the $\mathcal{O}((d_{\max} + 1) \ln T)$ regret bound can sometimes be much worse than $\mathcal{O}(\sqrt{T + d_{\text{tot}}})$. This occurs in scenarios when even a single round of feedback is delayed by $\Theta(T)$ rounds (e.g., missing feedback), undermining the benefits of having both regret guarantees under stronger curvature assumptions. Furthermore, to the best of our knowledge, no prior work has investigated whether improved regret guarantees are achievable for exp-concave losses under delayed feedback, leaving an important gap in the literature.

**Contribution.** To address these gaps, we propose a suite of algorithms and offer a comprehensive analysis for OCO with delayed feedback under both strongly convex and exp-concave losses, and we include a special case of (unconstrained) online linear regression with delays. The main contributions of this work can be summarized as follows (see also Table 1):

- We first consider the class of strongly convex losses in Section 3. Specifically, we propose an algorithm based on the follow-the-regularized-leader framework and obtain a $\mathcal{O}\left(\min\left\{\sigma_{\max}\ln T, \sqrt{d_{\text{tot}}}\right\} + \ln T\right)$ regret, where $\sigma_{\max}$ is the maximum number of missing observations over rounds. Compared with the results obtained by Wan et al. (2022a) and Wu et al. (2024), our results have several advantages. First, since $\sigma_{\max}$ is always no larger than $d_{\max}$ and can be significantly smaller than it, our $\sigma_{\max}\ln T$ bound improves upon the $d_{\max}\ln T$ bound in Wan et al. (2022a) and Wu et al. (2024). Second, we prove that our algorithm *simultaneously* achieves a $\mathcal{O}\left(\sqrt{d_{\text{tot}}} + \ln T\right)$ regret bound, making our algorithm no worse than the bound achieved by DOGD (Quanrud & Khashabi, 2015) either. Third, compared with the regret bounds obtained in Wan et al. (2022a) and Wu et al. (2024), our regret guarantee *does not depend on* the diameter of the action domain and recovers the one proven in Hazan et al. (2007) when there is no delay. Additionally, we provide a novel and improved analysis of the OMD-based algorithm originally proposed by Wu et al. (2024) in Appendix E, obtaining a regret bound that is again independent of the diameter of the action domain.

- In Section 4, we consider exp-concave losses, a broader function class compared to the strongly convex one. Specifically, we propose an algorithm based on the Online Newton Step (ONS) method that achieves a $\mathcal{O}\left(\min\left\{d_{\max}n\ln T, \sqrt{d_{\text{tot}}}\right\} + n\ln T\right)$ regret bound. To the best of our knowledge, this is the first algorithm to achieve logarithmic regret for exp-concave losses under delayed feedback, answering an open question proposed in Wan et al. (2022a). While both the bounds $d_{\max}n\ln T$ and $\sqrt{d_{\text{tot}}}$ can be achieved using a simple learning rate within the ONS framework, it is essential to use a delay-adaptive learning rate tuning scheme to achieve the best of these two guarantees within our analysis.

- In Section 5, we investigate online linear regression (OLR) problem, where the feasible domain is *unconstrained*, i.e., it corresponds to the entire $n$-dimensional Euclidean space $\mathbb{R}^n$. Leveraging the specific structure in OLR, we develop an algorithm based on the Vovk-Azoury-Warmuth forecaster, achieving a regret bound of

$\mathcal{O}\left(\|u\|_2^2(\min\left\{d_{\max}n\ln T, \sqrt{d_{\text{tot}}}\right\} + n\ln T)\right)$ without requiring any prior knowledge of neither the comparator $u \in \mathbb{R}^n$ nor the data. This result is achieved by incorporating a carefully designed clipping technique and, once again, employing an adaptive tuning of the learning rate.

- Finally, in Section 6, we implement all our proposed algorithms and conduct experiments to validate our theoretical results across multiple delayed settings and loss functions with different curvature properties. We also compare our methods with existing approaches to demonstrate their effectiveness.

## 1.1. Related works

**Online learning with curved losses.** While Abernethy et al. (2008) have shown that $\Theta(\sqrt{T})$ is the minimax regret for OCO, if the loss functions further enjoy curvature, the minimax regret can be improved. Hazan et al. (2007) show that OGD with a specific choice of learning rate achieves $\mathcal{O}(\frac{G^2}{\lambda}\ln T)$ regret for strongly convex losses where $G$ is the maximum $\ell_2$ norm of any loss gradient and $\lambda$ is the strong convexity parameter.[1] This upper bound is also minimax optimal as proven in Abernethy et al. (2008). For exp-concave losses, Hazan et al. (2007) proposed Online Newton Step (ONS) achieving $\mathcal{O}((\frac{1}{\alpha} + GD)\ln T)$ regret where $\alpha$ is the exp-concavity parameter and $D$ is the diameter of the feasible domain. Hazan et al. (2007) also proposed Exponential Weight Online Optimization (EWOO), achieving diameter and gradient scale independent guarantees. However, the algorithm is less practical due to its sampling complexity. For OLR, Vovk (2001) and Azoury & Warmuth (2001) independently introduced the Vovk-Azoury-Warmuth (VAW) forecaster achieving $\mathcal{O}(\ln T)$ regret without requiring prior knowledge of the data and the comparator. For a more detailed survey on OCO, we recommend the reader to Hazan (2016) and Orabona (2025).

**Online learning with delayed feedback.** Weinberger & Ordentlich (2002) initiated the study of online learning with delayed feedback, proposing an algorithm achieving $d \cdot R(T/d)$ where $d$ is the *fixed and known per-round delay* and $R(T)$ is the regret upper bound for some base algorithm that assumes no delay in the feedback. Specifically, their meta-algorithm runs $d + 1$ independent copies of the base algorithm on disjoint time lines in a round-robin fashion. However, this meta-algorithm is computationally expensive and does not show good empirical performances. Subsequently, Langford et al. (2009) proposed a practical algorithm by simply performing the gradient descent step using the observed gradients at each round, and achieved $\mathcal{O}(\sqrt{dT})$ and $\mathcal{O}(d\ln T)$ regret bounds for convex and strongly convex

---

[1]The definitions of these parameters are deferred to Section 2.

| Loss type | Regret bound | | |
|---|---|---|---|
| | Quanrud & Khashabi (2015) | Wan et al. (2022a); Wu et al. (2024) | Our work |
| Strongly convex | $\sqrt{d_{\text{tot}} + T}$ | $(d_{\max} + 1) \ln T$ | $\min\{\sigma_{\max} \ln T, \sqrt{d_{\text{tot}}}\} + \ln T$ |
| Exp-concave | $\sqrt{d_{\text{tot}} + T}$ | N/A | $\min\{d_{\max} n \ln T, \sqrt{d_{\text{tot}}}\} + n \ln T$ |
| Online linear regression | N/A | N/A | $\min\{d_{\max} n \ln T, \sqrt{d_{\text{tot}}}\} + n \ln T$ |

*Table 1.* Main results and comparisons with prior work. Here $T$ is the number of rounds, $n$ is the dimension of the feasible domain, $d_{\max}$ is the maximum delay, $\sigma_{\max} \le d_{\max}$ is the maximum number of missing observations, and $d_{\text{tot}}$ is the total delay. In Table 1, we omit the dependency on the curvature parameters, Lipschitz parameters, the norm of the comparator and domain diameter for conciseness. The detailed dependencies are explicitly shown in the respective theorem statements.

functions, respectively.

When delay is not uniform, Joulani et al. (2013) proposed BOLD (Black-box Online Learning with Delays) extending the method of Weinberger & Ordentlich (2002) and achieve $d_{\max} \cdot R(T/d_{\max})$ regret, but the algorithm still maintains multiple instances of base algorithms, which could be prohibitive in terms of computational costs. For convex functions, Quanrud & Khashabi (2015) achieved $\mathcal{O}(\sqrt{d_{\text{tot}}})$ where $d_{\text{tot}}$ is the total delay accumulated over $T$ rounds. Wan et al. (2022b; 2023) proposed a first Frank-Wolfe-type online algorithm to handle delayed feedback and obtain a regret bound of $\mathcal{O}(T^{3/4} + d_{\text{tot}}T^{-3/4})$ for general convex loss and $\mathcal{O}(T^{2/3} + d_{\max} \ln T)$ under strong convexity. There is also an interesting line of works whose focus is to obtain adaptive regret guarantees with delayed feedback (McMahan & Streeter, 2014; Joulani et al., 2016; Flaspohler et al., 2021) or to consider variants of delayed feedback (Gatmiry & Schneider, 2024; Bar-On & Mansour, 2025; Ryabchenko et al., 2025).

Two most related works to ours are Wan et al. (2022a) and Wu et al. (2024), which consider strongly convex losses together with delays. Specifically, Wan et al. (2022a) first proposed DOGD-SC for strongly convex losses, and establish a regret bound of $\mathcal{O}(\frac{GD+G^2}{\lambda} d_{\max} \ln T)$. Subsequently, Wu et al. (2024) proposed SDMD-RSC and obtained a $\mathcal{O}(\frac{d_{\max} G^2}{\lambda^2} + \frac{G^2+D}{\lambda} d_{\max} \ln T)$ regret bound.[2]

Beyond full-gradient feedback, there exists a growing interest in developing algorithms with delayed bandit feedback for a range of problems, including multi-armed bandits (Cesa-Bianchi et al., 2016; Cella & Cesa-Bianchi, 2020; Zimmert & Seldin, 2020; Masoudian et al., 2022; Van der Hoeven & Cesa-Bianchi, 2022; Esposito et al., 2023; Van der Hoeven et al., 2023; Masoudian et al., 2024; Schlisselberg et al., 2025; Zhang et al., 2025), Markov decision processes (Lancewicki et al., 2022; Jin et al., 2022; Van der Hoeven et al., 2023), and online convex optimization (Héliou et al., 2020; Bistritz et al., 2022; Wan et al., 2024).

---

[2]Wu et al. (2024) also considers the class of relative strongly convex loss functions.

## 2. Problem setting

Let $T \in \mathbb{N}$ be the time horizon and $n \in \mathbb{N}$ be the dimension. Denote by $\mathcal{X} \subset \mathbb{R}^n$ the domain, which we assume to be closed and non-empty. In each round $t \in [T]$, the learner selects a point $x_t \in \mathcal{X}$ as its decision and incurs a loss $f_t(x_t)$ given by some unknown function $f_t \colon \mathcal{X} \to \mathbb{R}$ that we assume to be convex and differentiable. Normally, in the standard OCO setting, the learner would then immediately observe the gradient $g_t = \nabla f_t(x_t)$. On the other hand, here we consider the delayed feedback scenario in which such a gradient $g_t$ is only observed at round $t + d_t$ with some *unknown arbitrary delay* $d_t \ge 0$. We assume $t + d_t \le T$ for all $t \in [T]$ without loss of generality (Joulani et al., 2013; 2016) because the feedback of any round $t$ with $t + d_t \ge T$ cannot be used the learner. The performance of the learner is then measured via the regret, which is defined as follows:

$$\text{Reg}_T = \max_{u \in \mathcal{X}} \text{Reg}_T(u) = \max_{u \in \mathcal{X}} \sum_{t=1}^{T} \big(f_t(x_t) - f_t(u)\big).$$

For convenience, we define $o_t = \{\tau \in \mathbb{N} : \tau + d_\tau < t\} \subseteq [t-1]$ to be the set of rounds whose gradients are observed before round $t$, and let $m_t = [t-1] \setminus o_t$ be the set of rounds whose observation is yet to be received at the beginning of round $t$. Define $\sigma_{\max} = \max_{t \in [T]} |m_t|$ to be the maximum number of missing observations over $T$ rounds, $d_{\max} = \max_{t \in [T]} d_t$ to be the maximum delay, and $d_{\text{tot}} = \sum_t d_t$ to be the total delay. Also define $d_{\max}^{\le t} = \max_{\tau \le t} \min\{d_\tau, t - \tau\}$ as the maximum delay that has been perceived up to round $t$.

Before presenting our main results, we must first introduce some definitions about the curvature of the loss functions.

**Definition 2.1.** A function $f \colon \mathcal{X} \to \mathbb{R}$ is $\lambda$-*strongly convex* with respect to $\|\cdot\|$ for $\lambda > 0$ if, for all $x, y \in \mathcal{X}$, $f(y) \ge f(x) + \langle \nabla f(x), y - x \rangle + \frac{\lambda}{2} \|y - x\|^2$.

**Definition 2.2.** A function $f \colon \mathcal{X} \to \mathbb{R}$ is $\alpha$-*exp-concave* for $\alpha > 0$ if $x \mapsto \exp(-\alpha f(x))$ is concave over $\mathcal{X}$.

We finally introduce some standard boundedness assumptions relating to the gradients and the domain.

**Algorithm 1** Delayed FTRL for strongly convex functions

**input** strong convexity parameter $\lambda > 0$
**initialize** $x_1 \in \mathcal{X}$
1: **for** $t = 1, 2, \ldots$ **do**
2:   Play $x_t$; receive $g_\tau = \nabla f_\tau(x_\tau)$ for all $\tau \in o_{t+1} \setminus o_t$
3:   Update

$$x_{t+1} = \arg\min_{x \in \mathcal{X}} \sum_{\tau \in o_{t+1}} \langle g_\tau, x \rangle + \frac{\lambda}{2} \sum_{s \leq t} \|x - x_s\|_2^2 \quad (1)$$

4: **end for**

**Assumption 2.3.** For every $t \in [T]$, the gradient of $f_t$ has norm bounded by $G \geq 0$, i.e., $\max_{x \in \mathcal{X}} \|\nabla f_t(x)\|_2 \leq G$.

**Assumption 2.4.** The diameter of $\mathcal{X}$ is bounded by $D \geq 0$, i.e., $\max_{x,y \in \mathcal{X}} \|x - y\|_2 \leq D$. We also assume $\mathbf{0} \in \mathcal{X}$.

**Other notations.** For a positive semidefinite matrix $A \in \mathbb{R}^{n \times n}$ and $x \in \mathbb{R}^d$, we denote $\|x\|_A = \sqrt{x^\top A x}$ to be the Mahalanobis norm induced by $A$ and, if $A$ is positive definite, let $\|x\|_{A^{-1}} = \sqrt{x^\top A^{-1} x}$ be the dual norm. We denote $\mathbf{1}$ as the all-one vector in an appropriate dimension.

# 3. Delayed OCO with strongly convex losses

In this section, we consider the problem of delayed OCO with strongly convex losses and propose Algorithm 1, which is built upon the follow-the-regularized-leader (FTRL) algorithm. Specifically, after receiving the gradients $g_\tau$ for all $\tau \in o_{t+1} \setminus o_t$ at the end of round $t$, we compute the updated decision $x_{t+1}$ as shown in Eq. (1), which is the minimizer of the cumulative linearized loss with respect to all the currently observed gradients, plus a squared $\ell_2$-regularization term with respect to *all the past decisions*. The following theorem shows that Algorithm 1 achieves $\mathcal{O}\big(\frac{G^2}{\lambda}\big(\ln T + \min\{\sigma_{\max}\ln T, \sqrt{d_{\text{tot}}}\}\big)\big)$ regret bound without any diameter assumption on the domain.

**Theorem 3.1.** *Assume that $f_1, \ldots, f_T$ are $\lambda$-strongly convex with respect to the Euclidean norm $\|\cdot\|_2$. Then, under Assumption 2.3, Algorithm 1 guarantees that*

$$\text{Reg}_T = \mathcal{O}\left(\frac{G^2}{\lambda}\left(\ln T + \min\left\{\sigma_{\max}\ln T, \sqrt{d_{\text{tot}}}\right\}\right)\right).$$

Theorem 3.1 highlights two advantages over previous works. From the perspective of the delay-related term, while both DOGD-SC (Wan et al., 2022a) and SDMD-RSC (Wu et al., 2024) achieve a $\mathcal{O}(d_{\max}\ln T)$ regret bound, the terms $\sigma_{\max}$ and $\sqrt{d_{\text{tot}}}$ in our regret bound can be substantially smaller than $d_{\max}$, with $\sigma_{\max} \leq d_{\max}$ always being true (Masoudian et al., 2022).[3] Second, while both DOGD-SC and

SDMD-RSC exhibit a polynomial dependence on the diameter $D$ of the action set $\mathcal{X}$, we remark that our bound *does not* depend on $D$ and recovers the optimal $\mathcal{O}\big(\frac{G^2}{\lambda}\ln T\big)$ regret in the no-delay setting.[4]

## 3.1. Regret analysis

Here we provide a proof sketch of Theorem 3.1, whereas the full proof is deferred to Appendix B. Specifically, using the strong convexity property, we first decompose the regret:

$$\text{Reg}_T(u) \leq \sum_{t=1}^{T}\left(\langle g_t, x_t - u\rangle - \frac{\lambda}{2}\|x_t - u\|_2^2\right)$$

$$= \underbrace{\sum_{t=1}^{T}\langle g_t, x_t^\star - u\rangle}_{\text{Reg}_T^\star(u)} + \underbrace{\sum_{t=1}^{T}\langle g_t, x_t - x_t^\star\rangle}_{\text{Drift}_T} - \frac{\lambda}{2}\sum_{t=1}^{T}\|x_t - u\|_2^2,$$

$$(2)$$

where $x_t^\star = \arg\min_{x \in \mathcal{X}} \sum_{\tau=1}^{t-1}(\langle g_\tau, x\rangle + \frac{\lambda}{2}\|x - x_\tau\|_2^2)$ for $t \geq 2$ and $x_1^\star = x_1$ are the decisions assuming that all gradients before round $t$ are observed.

Next, we analyze the term $\text{Reg}_T^\star(u)$ and $\text{Drift}_T$ separately. For the term $\text{Drift}_T$, applying the Cauchy-Schwarz inequality and using the fact that $\|g_t\|_2 \leq G$ for all $t \in [T]$ by Assumption 2.3, we can obtain that

$$\text{Drift}_T \leq G\sum_{t=1}^{T}\|x_t^\star - x_t\|_2 . \quad (3)$$

For the term $\text{Reg}_T^\star(u)$, following a standard FTRL analysis and using the optimality of $x_t^\star$, we are able to obtain that

$$\text{Reg}_T^\star(u) \leq \frac{\lambda}{2}\sum_{t=1}^{T}\|x_t - u\|_2^2 + \sum_{t=1}^{T}\langle g_t, x_t^\star - x_{t+1}^\star\rangle .$$

Since the first term can be canceled by the last negative term shown in Eq. (2), we only need to control the second term $\langle g_t, x_t^\star - x_{t+1}^\star\rangle$, which is further bounded by $G\|x_t^\star - x_{t+1}^\star\|_2$ via Cauchy-Schwarz and the fact that $\|g_t\|_2 \leq G$. Then, using a stability lemma for FTRL (Lemma A.2), we can show that

$$\|x_t^\star - x_{t+1}^\star\|_2 \leq \frac{2G}{\lambda(2t-1)} + \|x_t^\star - x_t\|_2 .$$

Interestingly, this inequality relates the Euclidean distance between adjacent "cheating" iterates $(x_t^\star)_{t \geq 1}$ in the stability term of FTRL to the distance between $x_t$ and $x_t^\star$, which is

---

[3]In fact, we also show in Lemma A.7 that $\sigma_{\max} \lesssim \sqrt{d_{\text{tot}}}$, and in Lemma A.9 that there are delay sequences such that $\sigma_{\max} \ll \sqrt{d_{\text{tot}}}$ and $\sigma_{\max} \approx \sqrt{d_{\text{tot}}}$, respectively.

[4]Despite the fact that $G$-Lipschitzness and $\lambda$-strong convexity of the losses over the domain $\mathcal{X}$ imply that its diameter is bounded by $2G/\lambda$, the guarantees of DOGD-SC and SDMD-RSC remain suboptimal.

also present in the $\mathtt{Drift}_T$ term and intuitively quantifies the influence of delays on the regret.

Combining the inequalities involving $\mathrm{Reg}_T^\star(u)$ and $\mathtt{Drift}_T$, we can finally bound the regret from above as follows:

$$
\begin{aligned}
\mathrm{Reg}_T &\leq \sum_{t=1}^{T} \frac{2G^2}{\lambda(2t-1)} + 2G \sum_{t=1}^{T} \|x_t^\star - x_t\|_2 \\
&\leq \frac{G^2}{\lambda} \ln(2T+1) + 2G \sum_{t=1}^{T} \|x_t^\star - x_t\|_2 .
\end{aligned}
$$

It remains to show how to bound $\|x_t^\star - x_t\|_2$ by $\mathcal{O}\big(\frac{G}{\lambda} \min\{\sigma_{\max} \ln T, \sqrt{d_{\mathrm{tot}}}\}\big)$, which is the key novelty in our analysis compared to previous works. Recalling the definitions of $x_t$ and $x_t^\star$, we can apply the stability lemma of FTRL (Lemma A.2) again and show for all $t \geq 2$ that

$$
\frac{\lambda(t-1)}{2} \|x_t^\star - x_t\|_2^2 \leq \frac{\left\|\sum_{\tau \in m_t} g_\tau\right\|_2^2}{2\lambda(t-1)} , \tag{4}
$$

meaning that $\sum_{t=1}^{T} \|x_t^\star - x_t\|_2 \leq \sum_{t=2}^{T} \frac{\left\|\sum_{\tau \in m_t} g_\tau\right\|_2}{\lambda(t-1)} \leq \sum_{t=2}^{T} \frac{G|m_t|}{\lambda(t-1)}$, where we also use the fact that $x_1^\star = x_1$. Here, we highlight the importance of including all previous decisions $x_\tau$ for $\tau \leq t$, instead of $\tau \in o_{t+1}$ only, in the regularization term of the update rule of $x_{t+1}$ shown in Eq. (1). Doing so particularly ensures that the updates of $x_t$ and $x_t^\star$ share the same regularization terms, which is crucial in leading to a diameter-free upper bound for $\|x_t^\star - x_t\|_2$ using the stability lemma.

Finally, we study the term $\sum_{t=2}^{T} \frac{|m_t|}{t-1}$. Directly bounding $|m_t|$ from above by $\sigma_{\max}$ leads to the first $\sigma_{\max} \ln T$ bound. To further obtain the $\sqrt{d_{\mathrm{tot}}}$ bound, it is crucial to observe that $\sum_{\tau \leq t} |m_\tau| \leq (t-1)^2$ since $m_\tau \subseteq [\tau - 1]$. Therefore, by also using Orabona (2025, Lemma 4.13) we are able to prove that $\sum_{t=2}^{T} \frac{|m_t|}{t-1} \leq \sum_{t=2}^{T} \frac{|m_t|}{\sqrt{\sum_{\tau \leq t} |m_\tau|}} \leq 2\sqrt{d_{\mathrm{tot}}}$, which concludes the regret analysis.

## 4. Delayed OCO with exp-concave losses

In this section, we consider the delayed OCO problem with exp-concave losses. Exp-concave losses are a more general class of loss functions that require more sophisticated techniques to be tackled. To address this problem, we design Algorithm 2, a variant of Online Newton Step (ONS) which effectively handles delayed feedback. Specifically, after receiving the gradients $g_\tau$ for all $\tau \in o_{t+1} \backslash o_t$, we select $x_{t+1}$ as the minimizer of the cumulative surrogate loss over all the already observed gradients and the past actions, with an additive squared $\ell_2$-regularization term. For simplicity, in this section we omit dependencies on curvature parameters, Lipschitz constants, and domain diameter; they appear explicitly in the theorem statements. The following result provides a first regret bound for Algorithm 2.

---

**Algorithm 2** Delayed ONS for exp-concave functions

**input** $\beta > 0$, learning rate rule $\{\eta_t\}_{t \geq 1}$,
**initialize** $x_1 \in \mathcal{X}$
1: **for** $t = 1, 2, \dots$ **do**
2:     Play $x_t$; receive $g_\tau = \nabla f_\tau(x_\tau)$ for all $\tau \in o_{t+1} \backslash o_t$
3:     $x_{t+1} = \arg\min_{x \in \mathcal{X}} \sum_{\tau \in o_{t+1}} \left( \langle g_\tau, x \rangle + \frac{\beta}{2} \langle g_\tau, x - x_\tau \rangle^2 \right)$
        $+ \frac{\eta_t}{2} \|x\|_2^2$
4: **end for**

---

**Theorem 4.1.** *Assume that $f_1, \dots, f_T$ are $\alpha$-exp-concave and let $\beta = \frac{1}{2} \min\{\frac{1}{4GD}, \alpha\}$. Then, under Assumptions 2.3 and 2.4, Algorithm 2 with $0 < \eta_0 \leq \eta_1 \leq \cdots \leq \eta_T$ guarantees that*

$$
\mathrm{Reg}_T = \mathcal{O}\Big( \frac{n}{\beta} \ln\Big(1 + \frac{\beta G^2 T}{\eta_0 n}\Big) + \eta_T D^2 + \min\{B_1, B_2\} \Big),
$$

*where $B_1 = \left(\frac{G^2}{\eta_0} + \frac{1}{\beta}\right) n d_{\max} \ln\left(1 + \frac{\beta G^2 T}{\eta_0 n}\right)$ and $B_2 = G^2 \sum_{t=1}^{T} \frac{|m_t|}{\eta_{t-1}}$.*

We can now introduce two careful tunings of the time-varying learning rates $(\eta_t)_{t \geq 1}$ to derive the regret bounds $\mathcal{O}(d_{\max} n \ln T)$ and $\mathcal{O}(\sqrt{d_{\mathrm{tot}}})$ individually.

**Simple tuning.** With a constant learning rate constant $\eta_t = 1$ for all $t \in [T]$, Algorithm 2 directly obtains $\mathcal{O}(d_{\max} n \ln T)$ regret. Alternatively, setting $\eta_t = \frac{G}{D} \sqrt{\sum_{s \leq t} |m_s| + |m_t| + 1}$ for all $t \geq 1$, Algorithm 2 achieves $\mathcal{O}(\sqrt{d_{\mathrm{tot}}})$ regret; here, the $|m_t| + 1$ term is an essentially tight worst-case estimation of $|m_{t+1}|$, since $m_{t+1} \subseteq m_t \cup \{t\}$.

Note that either of these two bounds can be significantly better than the other under different delay sequences, e.g., as shown by our Lemma A.10 in the appendix. Therefore, we ideally want to achieve $\mathcal{O}(\min\{d_{\max} n \ln T, \sqrt{d_{\mathrm{tot}}}\})$ regret via a single choice of the learning rates. In fact, we can show that it is indeed possible to obtain such a bound by a careful delay-adaptive learning rate tuning.

**Adaptive tuning.** The adaptive learning rate is given by $\eta_0 = 1$ and $\eta_t = \min\{a_t, b_t\} + 1$ for all $t \geq 1$, where

$$
a_t = \frac{2}{GD} \left( G^2 + \frac{1}{\beta} \right) n d_{\max}^{\leq t} \ln\left( 1 + \frac{\beta G^2 T}{n} \right) , \tag{5}
$$

$$
b_t = \frac{G}{D} \sqrt{\sum_{s \leq t} |m_s| + |m_t| + 1} . \tag{6}
$$

The overall idea behind this learning rate tuning is to keep track of both the $d_{\max} n \ln T$ and $\sqrt{d_{\mathrm{tot}}}$ regret guarantees over the rounds via $a_t$ and $b_t$, respectively. Then, $\eta_t$ is set

depending on the best of the two, i.e., $\min\{a_t, b_t\}$, which then leads to achieve the best of both regret bounds. Note that this adaptive tuning requires the knowledge of the time-stamps of the received gradients since we need to compute $d_{\max}^{\leq t} = \max_{\tau \leq t} \min\{d_\tau, t - \tau\}$ which, we recall, is the maximum delay that has been perceived up to round $t$. The following corollary provides a regret bound for Algorithm 2 with this adaptive tuning. The full proof of Corollary 4.2 can be found in Appendix C.

**Corollary 4.2.** *Assume that $f_1, \ldots, f_T$ are $\alpha$-exp-concave and let $\beta = \frac{1}{2} \min\{\frac{1}{4GD}, \alpha\}$. Then, under Assumptions 2.3 and 2.4, Algorithm 2 with the adaptive learning rate $\eta_t = \min\{a_t, b_t\} + 1$, where $a_t$ and $b_t$ are defined in Equations (5) and (6), guarantees that*

$$\text{Reg}_T = \mathcal{O}\left(\frac{n}{\beta} \ln\left(1 + \frac{\beta G^2 T}{n}\right) + D^2 + \min\{C_1, C_2\}\right),$$

*where $C_1 = \left(\frac{D}{G} + 1\right)\left(G^2 + \frac{1}{\beta}\right) n d_{\max} \ln\left(1 + \frac{\beta G^2 T}{n}\right)$ and $C_2 = \left(G^2 + GD\right)\left(\sqrt{d_{\text{tot}}} + 1\right)$.*

Corollary 4.2 shows Algorithm 2 with the adaptive learning rate obtains regret $\mathcal{O}\left(\min\left\{d_{\max} n \ln T, \sqrt{d_{\text{tot}}}\right\}\right)$. The main advantage of an adaptive learning rate is that it requires no prior knowledge of $d_{\text{tot}}$ or $d_{\max}$, nor does it rely on a doubling trick that would throw away information via resets.

### 4.1. Regret analysis

In this section, we provide a proof sketch of Theorem 4.1 and Corollary 4.2, while their full proofs are deferred to Appendix C. Specifically, using the exp-concavity property and Lemma A.3, we decompose the overall regret as follows:

$$\text{Reg}_T(u) \leq \underbrace{\sum_{t=1}^{T} \langle g_t, x_t^\star - u \rangle}_{\text{Reg}_T^\star(u)} + \underbrace{\sum_{t=1}^{T} \langle g_t, x_t - x_t^\star \rangle}_{\text{Drift}_T}$$
$$- \frac{\beta}{2} \sum_{t=1}^{T} \langle g_t, x_t - u \rangle^2, \quad (7)$$

where we define $x_1^\star = x_1$ and, for $t \geq 2$, $x_t^\star = \arg\min_{x \in \mathcal{X}} \sum_{\tau=1}^{t-1} (\langle g_\tau, x \rangle + \frac{\beta}{2} \langle g_\tau, x - x_\tau \rangle^2) + \frac{\eta_{t-1}}{2} \|x\|_2^2$ to be the decisions assuming that all gradients before round $t$ are observed.

For the term $\text{Reg}_T^\star(u)$, following a standard FTRL analysis, we are able to obtain that

$$\text{Reg}_T^\star(u) \leq \frac{\eta_T}{2} \|u\|_2^2 + \frac{\beta}{2} \sum_{t=1}^{T} \langle g_t, u - x_t \rangle^2$$
$$+ \sum_{t=1}^{T} \min\left\{GD, \|g_t\|_{A_{t-1}^{-1}}^2\right\}. \quad (8)$$

where $A_{t-1} = \eta_{t-1} I + \beta \sum_{\tau=1}^{t-1} g_\tau g_\tau^\top$. Applying Lattimore & Szepesvári (2020, Lemma 19.4), the last sum on the right-hand side of the above inequality satisfies

$$\sum_{t=1}^{T} \min\left\{GD, \|g_t\|_{A_{t-1}^{-1}}^2\right\} = \mathcal{O}\left(\frac{n}{\beta} \ln\left(1 + \frac{\beta G^2 T}{n}\right)\right). \quad (9)$$

Now we consider the $\text{Drift}_T$ term. By applying the Cauchy-Schwarz inequality followed by the stability lemma (Lemma A.2) again, it follows that for all $t \geq 1$,

$$\text{Drift}_T \leq \sum_{t=1}^{T} \|g_t\|_{A_{t-1}^{-1}} \|x_t - x_t^\star\|_{A_{t-1}}$$
$$\leq 4 \sum_{t=1}^{T} \|g_t\|_{A_{t-1}^{-1}} \left(\sum_{\tau \in m_t} \|g_\tau\|_{A_{t-1}^{-1}}\right). \quad (10)$$

By applying Lemma C.1, it holds that

$$\text{Drift}_T = \mathcal{O}\left(\left(G^2 + \frac{1}{\beta}\right) n d_{\max} \ln\left(1 + \frac{\beta G^2 T}{n}\right)\right). \quad (11)$$

At the same time, we can also prove that

$$\text{Drift}_T = \mathcal{O}\left(G^2 \sum_{t=1}^{T} \frac{|m_t|}{\eta_{t-1}}\right). \quad (12)$$

Combing Equations (7) to (12) concludes the proof of Theorem 4.1. To prove Corollary 4.2, we carefully consider the adaptive learning rate tuning and separate the analysis into two cases. In case $a_T \leq b_T$ at the end of the $T$ rounds, we utilize a delayed version of the elliptical potential lemma (Lemma C.1) to achieve the logarithmic regret. On the other hand, if $b_T < a_T$ we split the regret analysis at the last round $\tau^\star$ at which $a_{\tau^\star} \leq b_{\tau^\star}$. Then, we use again the logarithmic bound up to round $\tau^\star$ and the $\sqrt{d_{\text{tot}}}$ bound for the remaining rounds. It suffices to observe that the first bound is no worse than $\sqrt{d_{\text{tot}}}$ since $a_{\tau^\star} \leq b_{\tau^\star}$ to conclude the proof.

## 5. Online linear regression with delayed labels

Here we consider the problem of online linear regression (OLR) with delays. This setting essentially corresponds to a variant of OCO where the domain is $\mathcal{X} = \mathbb{R}^n$ and loss functions are $f_t(x) = \frac{1}{2}(\langle z_t, x \rangle - y_t)^2$ comparing any point $x \in \mathbb{R}^n$ to a label $y_t \in \mathbb{R}$ given some feature vector $z_t \in \mathbb{R}^n$; to be precise, the predicted label by a given point $x$ corresponds to the inner product $\langle z_t, x \rangle$. At each round $t$, the learner first observes an $n$-dimensional feature vector $z_t$ before performing its prediction $x_t$, but the true label $y_t$ is only revealed at a later round $t + d_t$. A common assumption on feature vectors and labels in this setting, analogous to

the ones we introduced in Section 2 for instance, is their boundedness.

**Assumption 5.1.** The feature vectors $z_1, \ldots, z_T$ and the labels $y_1, \ldots, y_T$ are bounded, i.e., $\|z_t\|_2 \leq Z$ and $|y_t| \leq Y$ for any $t \in [T]$, given $Y, Z \geq 0$.

---
**Algorithm 3** Delayed VAW forecaster with clipping
---
**input** learning rate rule $\{\eta_t\}_{t \geq 1}$
**initialize** $\rho_1 = 0$
1: **for** $t = 1, 2, \ldots$ **do**
2:      Observe $z_t$
3:      Set $x_t = \arg\min_{x \in \mathbb{R}^n} \sum_{\tau \in o_t} -y_\tau \langle z_\tau, x \rangle + \frac{\eta_t}{2}\|x\|_2^2$
                     $+\frac{1}{2}\sum_{\tau \leq t}\langle z_\tau, x \rangle^2$
4:      Play $\widetilde{x}_t = x_t \cdot \min\left\{\frac{\rho_t}{|\langle z_t, x_t \rangle|}, 1\right\}$
5:      Receive $y_\tau$ for all $\tau \in o_{t+1} \setminus o_t$
6:      Set $\rho_{t+1} = \max_{\tau \in o_{t+1}} |y_\tau|$
7: **end for**
---

Note that the loss $f_t$ becomes exp-concave when the domain is also *bounded*. If this were the case, we could solve this problem by designing a version of ONS that can handle delayed labels. In OLR, however, the domain is unconstrained as it corresponds to the whole $n$-dimensional Euclidean space, which makes it particularly challenging to simply adapt one of the techniques seen so far without further assumptions. We instead design an algorithm for this problem (see Algorithm 3) that corresponds to an adaptation of the Vovk-Azoury-Warmuth (VAW) forecaster (Azoury & Warmuth, 2001; Vovk, 2001) in order to handle delayed labels. We can then prove that the regret guarantee for this algorithm in the delayed OLR setting becomes as stated in Theorem 5.2 below (whose proof is in Appendix D).

**Theorem 5.2.** *In the OLR problem with delayed labels under Assumption 5.1, Algorithm 3 guarantees for any $0 < \eta_0 \leq \eta_1 \leq \cdots \leq \eta_T$ that*

$$\text{Reg}_T(u) \leq \frac{\eta_T}{2}\|u\|_2^2 + nY^2 \ln\left(1 + \frac{Z^2 T}{\eta_0 n}\right)$$
$$+ \mathcal{O}\left(Y^2\big(\sigma_{\max} + \min\{M_1, M_2\}\big)\right),$$

*where* $M_1 = nd_{\max} \ln\big(1 + \frac{Z^2 T}{\eta_0 n}\big)$ *and* $M_2 = Z^2 \sum_{t=1}^T \frac{|m_t|}{\eta_t}$.

The idea behind the regret analysis is once again to decompose the regret into a cheating regret term and a drift term:

$$\text{Reg}_T(u) = \underbrace{\sum_{t=1}^T \big(f_t(x_t^\star) - f_t(u)\big)}_{\text{Reg}_T^\star(u)} + \underbrace{\sum_{t=1}^T \big(f_t(\widetilde{x}_t) - f_t(x_t^\star)\big)}_{\text{Drift}_T},$$

where $(\widetilde{x}_t)_{t \geq 1}$ are the actions played by Algorithm 3, while $(x_t^\star)_{t \geq 1}$ are the "cheating" iterates that assume to have

knowledge about all labels from past rounds. To bound the cheating regret $\text{Reg}_T^\star(u)$, it is important to leverage the curvature of the squared loss. Specifically, by definition,

$$\text{Reg}_T^\star(u) = \sum_{t=1}^T \frac{\langle z_t, x_t^\star \rangle^2 - \langle z_t, u \rangle^2}{2} + \sum_{t=1}^T \langle -y_t z_t, x_t^\star - u \rangle.$$

Then, we can study the second sum via the standard tools for the regret analysis of FTRL with respect to the linear losses $x \mapsto -y_t \langle z_t, x \rangle$, which yields

$$\text{Reg}_T^\star(u) \leq \frac{\eta_T}{2}\|u\|_2^2 + nY^2 \ln\left(1 + \frac{Z^2 T}{\eta_0 n}\right).$$

This is exactly the first line in the regret guarantee presented in Theorem 5.2, and it corresponds to the part that does not depend on delays.

On the other hand, the drift term $\text{Drift}_T$ requires much more care and novel techniques. By the convexity of $f_t$, we have that $\text{Drift}_T \leq \sum_{t=1}^T \langle \nabla f_t(\widetilde{x}_t), \widetilde{x}_t - x_t^\star \rangle$. Here we immediately observe the importance of the additional clipping of $x_t$ to define the selected point $\widetilde{x}_t$, which is inspired from the clipping ideas by Cutkosky (2019); Mayo et al. (2022). Its scope is to guarantee that the predicted label $\langle z_t, \widetilde{x}_t \rangle$ falls within the range of true labels; the reason for this is to avoid the gradient of $f_t$ evaluated at $\widetilde{x}_t$ to blow up, otherwise obstructing an attempt to nicely bound $\text{Drift}_T$. We also remark that, differently form Mayo et al. (2022), we do not require to clip the labels used in the iterates update too. If we had knowledge of $Y$, we could use it to clip to the interval $[-Y, Y]$, thus guaranteeing $f_t(\widetilde{x}_t) \leq Y$. However, since we want to assume *no prior knowledge of $Y$*, the best clipping we can do at any time $t$ is via $\rho_t$. Doing so requires to handle possible rounds when the label falls outside the clipping interval, which in turn requires a careful analysis that accounts for the feedback to be revealed only after some delay (as $\rho_t$ could possibly be updated much later in time). We are then able to prove that

$$\text{Drift}_T = \mathcal{O}\big(Y^2 \sigma_{\max} + Y^2 \min\{M_1, M_2\}\big).$$

which is the delay-dependent part of the regret; the $Y^2 \sigma_{\max}$ term, in particular, is the one due to clipping mistakes.

Given any $\gamma > 0$, we may now set $\eta_0 = \gamma$ and $\eta_t = \gamma(\min\{a_t, b_t\} + 1)$ for all $t \geq 1$, where

$$a_t = 2nd_{\max}^{\leq t} \ln\left(1 + \frac{Z^2 T}{\gamma n}\right), \quad b_t = Z\sqrt{\sum_{s \leq t}|m_s|}.$$

(13)

By doing so, we obtain the following Corollary 5.3 which provides a regret bound for Algorithm 3 with this adaptive tuning, and whose proof is deferred to Appendix D.

**Corollary 5.3.** *In the OLR problem with delayed labels under Assumption 5.1, Algorithm 3 with the adaptive learning*

*rate $\eta_t = \gamma(\min\{a_t, b_t\} + 1)$, where $a_t$ and $b_t$ are defined in Equation (13) for any $\gamma > 0$ guarantees that*

$$\text{Reg}_T \leq \frac{\gamma\|u\|_2^2}{2} + nY^2 \ln\left(1 + \frac{Z^2 T}{\gamma n}\right) + \mathcal{O}\left(\min\{Q_1, Q_2\}\right),$$

*where $Q_1 = \left(\gamma\|u\|_2^2 + Y^2\right) n d_{\max} \ln\left(1 + \frac{Z^2 T}{\gamma n}\right)$ and $Q_2 = \left(\gamma Z\|u\|_2^2 + (Z+1)Y^2\right)\sqrt{d_{\text{tot}}}$.*

To achieve this final result, we leverage similar ideas from the adaptive tuning for delayed ONS in Corollary 4.2, as mentioned above, together with a nontrivial relation between $\sigma_{\max}$ and $\sqrt{d_{\text{tot}}}$ to handle the additive $Y^2\sigma_{\max}$ term from the clipping errors (see Lemma A.7). We remark that here we used directly $Z$ for the tuning, which requires its knowledge since the first round; we could easily do without this prior knowledge by using $Z_t = \max_{\tau \leq t}\|z_\tau\|_2$ instead because we always observe all the previous and the current feature vectors by the beginning of round $t$.

## 6. Experiments

In this section, we evaluate the performance of the proposed algorithms on three types of loss functions in the delayed OCO setting.[5] All experiments are conducted over $T = 10000$ round and results are averaged over 20 independent trials. To showcase the advantage of our algorithms, we consider two delay regimes. For the first case, each delay $d_t$ is independently and uniformly sampled from the set $\{0, 1, \ldots, 5\}$, thus leading to $\mathbb{E}[\sqrt{d_{\text{tot}}}] = \Theta(\sqrt{T})$ and $\mathbb{E}[\sigma_{\max}] \leq \mathbb{E}[d_{\max}] \leq 5$. In the second case, we define $p = T^{-1/3} = 0.1$. Then, for each $t$, $d_t$ is sampled from the same distribution with probability $1 - p$, and it is set to be $d_t = T - t$ with probability $p$. In this case, $\mathbb{E}[\sqrt{d_{\text{tot}}}] = o(T)$, $\mathbb{E}[d_{\max}] \geq T(1 - (1-p)^T)$, and $\mathbb{E}[\sigma_{\max}] = \mathcal{O}(pT)$. We compare our algorithms against several baselines designed for delayed feedback settings. Below, we describe how we construct losses, together with the baseline algorithms we compare against. We provide additional experiments in Appendix F.

**Strongly convex loss.** We consider the following strongly convex losses $f_t(x) = \frac{1}{2}(\langle z_t, x \rangle - y_t)^2 + \frac{1}{2}\|x\|_2^2$. The feasible set is the ball $\mathcal{X} = \{x \in \mathbb{R}^5, \|x\|_2 \leq 2\}$. Each coordinate of the feature vector $z_t \in \mathbb{R}^5$ at round $t$ is uniformly chosen from $[-1, 1]$ while $y_t = \langle z_t, \mathbf{1}\rangle + \epsilon_t$, where $\epsilon_t$ is an i.i.d. standard Gaussian noise. We evaluate Algorithm 1 on this loss sequence and compare its performance with DOGD-SC (Wan et al., 2022a), SDMD-RSC (Wu et al., 2024, Algorithm 6), and BOLD-OGD which applies the reduction proposed by Joulani et al. (2013) to OGD.

**Exp-concave loss.** The loss functions we consider for exp-concave ones are $f_t(x) = \frac{1}{2}(\langle z_t, x \rangle - y_t)^2$. The other configurations are the same as the experiments in the strongly convex case. We evaluate our Algorithm 2 and compare its performance with that of DOGD (Quanrud & Khashabi, 2015) and BOLD-ONS, which applies the reduction proposed in Joulani et al. (2013) to ONS (Hazan et al., 2007).

**Online linear regression.** We still consider the loss function $f_t(x) = \frac{1}{2}(\langle z_t, x \rangle - y_t)^2$ for all $t \in [T]$, the same one as used in the exp-concave setting. The only difference is that the action space is now unconstrained ($\mathcal{X} = \mathbb{R}^5$). We empirically evaluate Algorithm 3 on this loss sequence and compare the performance with DOGD (Quanrud & Khashabi, 2015) and BOLD-VAW, which is again a combination of the reduction in Joulani et al. (2013) and the VAW forecaster (Azoury & Warmuth, 2001; Vovk, 2001).

**Experimental results.** Figure 1 shows the mean cumulative regret and its standard deviation over 20 rounds for the instances with strong convexity, exp-concavity, and OLR under the two previously mentioned delay regimes. For strongly convex losses, we find that our algorithm performs much better than DOGD-SC (Wan et al., 2022a) and have similar performances compared to SDMD-RSC, which is proven to only achieve $\mathcal{O}(d_{\max} \ln T)$ regret (Wu et al., 2024). However, we point out that this mismatch in the empirical performance and the theoretical guarantee of SDMD-RSC is due to a loose analysis of this algorithm. In fact, we show that SDMD-RSC can also achieve the same $\mathcal{O}(\min\{\sigma_{\max}\ln T, \sqrt{d_{\text{tot}}}\})$ regret via a refined analysis. The proof is deferred to Appendix E.

For both exp-concave and OLR settings, our algorithms consistently outperform DOGD, which does not leverage the curvature of the loss function, as well as the reduction-based algorithms proposed in Joulani et al. (2013), under both delay regimes, showing the effectiveness of our algorithms under different delay conditions.

## 7. Conclusions

In this paper, we study how to leverage the curvature of the loss functions in online convex optimization with delayed feedback so as to improve regret guarantees. For strongly convex functions, we derive an algorithm achieving $\mathcal{O}(\min\{\sigma_{\max}\ln T, \sqrt{d_{\text{tot}}}\})$ regret, improving upon previous work (Wan et al., 2022a; Wu et al., 2024), which only obtain $\mathcal{O}(d_{\max}\ln T)$ regret. We also derive $\mathcal{O}(\min\{d_{\max} n \ln T, \sqrt{d_{\text{tot}}}\})$ for exp-concave losses and online linear regression, answering an open question proposed in Wan et al. (2022a). It is still left open whether $\mathcal{O}(\min\{\sigma_{\max} n \ln T, \sqrt{d_{\text{tot}}}\})$ is achievable for exp-concave losses.

---

[5]The code for the experiments is available at https://github.com/haoqiu95/DOCO.

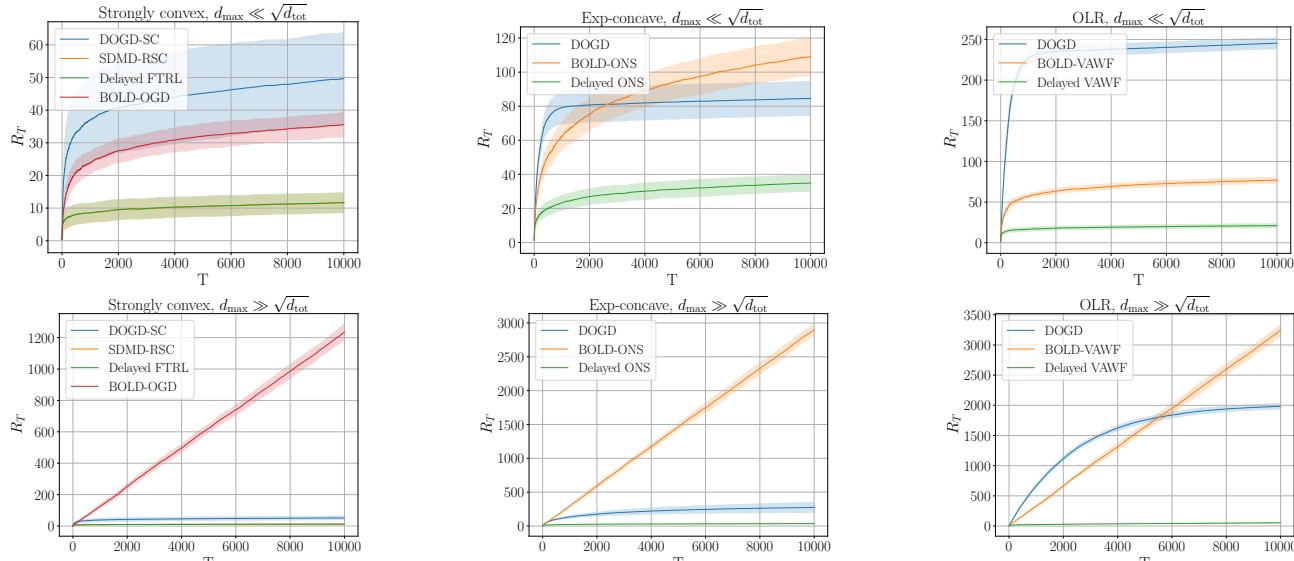

*Figure 1.* Comparison with relevant baselines. The shaded areas consider a range centered around the mean with half-width corresponding to the empirical standard deviation over 20 repetitions.

## Acknowledgements

EE and HQ acknowledge the financial support from the FAIR (Future Artificial Intelligence Research) project, funded by the NextGenerationEU program within the PNRR-PE-AI scheme (M4C2, investment 1.3, line on Artificial Intelligence), the EU Horizon CL4-2022-HUMAN-02 research and innovation action under grant agreement 101120237, project ELIAS (European Lighthouse of AI for Sustainability), and the One Health Action Hub, University Task Force for the resilience of territorial ecosystems, funded by Università degli Studi di Milano (PSR 2021-GSA-Linea 6).

## Impact Statement

This paper presents work whose goal is to advance the field of Machine Learning. The contribution of our work is primarily theoretical in nature. Thus, we do not foresee any societal consequences.

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

# A. Auxiliary results

In this section, we show several auxiliary lemmas that will be helpful throughout the paper.

## A.1. General results for the regret analysis

The following lemma is a standard result for the regret of FTRL.

**Lemma A.1** (Orabona (2025, Lemma 7.1)). *Let $\mathcal{X} \subseteq \mathbb{R}^n$ be closed and non-empty. Denote by $F_t(x) = \psi_t(x) + \sum_{\tau=1}^{t-1} \ell_\tau(x)$. Assume that $\arg\min_{x \in \mathcal{X}} F_t(x)$ is not empty and $x_t \in \arg\min_{x \in \mathcal{X}} F_t(x)$. Then, for any $u \in \mathcal{X}$,*

$$\sum_{t=1}^T \big(\ell_t(x_t) - \ell_t(u)\big) = \psi_{T+1}(u) - \min_{x \in \mathcal{X}} \psi_1(x) + \sum_{t=1}^T \big[F_t(x_t) - F_{t+1}(x_{t+1}) + \ell_t(x_t)\big] + F_{T+1}(x_{T+1}) - F_{T+1}(u) .$$

The next lemma bounds the distance between two FTRL iterates with different linear losses and possibly different regularizers. It also shows a simplified upper bound in the case when the two iterates have the same regularizer.

**Lemma A.2** (Stability lemma). *Let $\mathcal{X} \subseteq \mathbb{R}^n$ be closed and non-empty. Let $A_1, A_2 \succeq 0$ be two positive semidefinite matrices, $b_1, b_2 \in \mathbb{R}^n$, and $c_1, c_2 \in \mathbb{R}$. Define $\psi_1(x) = x^\top A_1 x + b_1^\top x + c_1$ and $\psi_2(x) = x^\top A_2 x + b_2^\top x + c_2$. Suppose that $z_1 \in \arg\min_{x \in \mathcal{X}} \{\langle w_1, x\rangle + \psi_1(x)\}$ and $z_2 \in \arg\min_{x \in \mathcal{X}} \{\langle w_2, x\rangle + \psi_2(x)\}$. Then, we have*

$$\|z_1 - z_2\|_{A_1}^2 + \|z_1 - z_2\|_{A_2}^2 \leq \langle w_1 - w_2, z_2 - z_1\rangle + (\psi_1(z_2) - \psi_2(z_2)) - (\psi_1(z_1) - \psi_2(z_1)) .$$

*Furthermore, if $\psi_1(x) = \psi_2(x) = x^\top A x + b^\top x + c$ with positive definite $A \succ 0$, we have*

$$\|z_1 - z_2\|_A \leq \frac{1}{2}\|w_1 - w_2\|_{A^{-1}} .$$

*Proof.* Let $h_1(x) = \langle w_1, x\rangle + \psi_1(x)$ and $h_2(x) = \langle w_2, x\rangle + \psi_2(x)$ be twice-differentiable functions with Hessians $A_1 + A_1^\top$ and $A_2 + A_2^\top$, respectively. Note that $z_1 \in \arg\min_{x \in \mathcal{X}} h_1(x)$ and $z_2 \in \arg\min_{x \in \mathcal{X}} h_2(x)$. By Taylor's theorem and first-order optimality conditions, we know that

$$(\langle w_1, z_2\rangle + \psi_1(z_2)) - (\langle w_1, z_1\rangle + \psi_1(z_1)) = h_1(z_2) - h_1(z_1) \geq \|z_1 - z_2\|_{A_1}^2 ,$$
$$(\langle w_2, z_1\rangle + \psi_2(z_1)) - (\langle w_2, z_2\rangle + \psi_2(z_2)) = h_2(z_1) - h_2(z_2) \geq \|z_1 - z_2\|_{A_2}^2 .$$

Summing up the above two inequalities, we obtain

$$\|z_1 - z_2\|_{A_1}^2 + \|z_1 - z_2\|_{A_2}^2 \leq \langle w_1 - w_2, z_2 - z_1\rangle + (\psi_1(z_2) - \psi_2(z_2)) - (\psi_1(z_1) - \psi_2(z_1)) .$$

The second result is directly obtained by applying the Cauchy-Schwarz inequality when $\psi_1(x) = \psi_2(x)$. $\qquad\square$

The following lemma is the quadratic bound of $\alpha$-exp-concave functions.

**Lemma A.3** (Hazan et al. (2007, Lemma 3)). *Let $f \colon \mathcal{X} \to \mathbb{R}$ be an $\alpha$-exp-concave function. Then, under Assumptions 2.3 and 2.4, we have that*

$$f(x) \geq f(y) + \langle \nabla f(y), x - y\rangle + \frac{\beta}{2}\big(\langle \nabla f(y), x - y\rangle\big)^2$$

*for any $x, y \in \mathcal{X}$, where $\beta = \frac{1}{2}\min\{\frac{1}{4GD}, \alpha\}$.*

The following lemma is the link of the Bregman divergences between 3 points.

**Lemma A.4** (Wei et al. (2021, Lemma 10)). *Let $\mathcal{A}$ be a convex set and $x_2 = \arg\min_{x \in \mathcal{A}} \{\langle g, x\rangle + D_\psi(x, x_1)\}$. Then, for any $u \in \mathcal{A}$,*

$$\langle x_2 - u, g\rangle \leq D_\psi(u, x_1) - D_\psi(u, x_2) - D_\psi(x_2, x_1) .$$

The following lemma is the general bound on $\langle g, v\rangle - \frac{\lambda}{2}\|v\|^2$, which related to the one achievable via the Fenchel-Young inequality but strengthened thanks to a norm constraint on $v$.

**Lemma A.5** (Flaspohler et al. (2021, Lemma 18)). *Let $\|\cdot\|$ be a norm over $\mathbb{R}^n$ and let $\|\cdot\|_*$ be its dual norm. For any constants $\lambda, c, b > 0$ and any $g \in \mathbb{R}^n$,*

$$\sup_{v \in \mathbb{R}^n : \|v\| \leq \min\{\frac{c}{\lambda}, b\}} \Big(\langle g, v\rangle - \frac{\lambda}{2}\|v\|^2\Big) \leq \min\Big\{\frac{1}{2\lambda}\|g\|_*^2, \frac{c}{\lambda}\|g\|_*, b\|g\|_*\Big\} .$$

### A.2. Results for delay-related quantities

The following three lemmas quantify the relationship between $\sigma_{\max}$, $d_{\max}$, and $d_{\text{tot}}$.

**Lemma A.6** (Masoudian et al. (2022, Lemma 3)). *Let $d_{\max}(S) = \max_{\tau \in S} d_\tau$ and $\bar{S} = [T] \setminus S$ for any $S \subseteq [T]$. Then,*

$$\sigma_{\max} \leq \min_{S \subseteq [T]} \left( |S| + d_{\max}(\bar{S}) \right) .$$

**Lemma A.7.** *Let $d_{\text{tot}}(S) = \sum_{\tau \in S} d_\tau$ and $\bar{S} = [T] \setminus S$ for any $S \subseteq [T]$. Then,*

$$\sigma_{\max} \leq 2\sqrt{2} \min_{S \subseteq [T]} \left( |S| + \sqrt{d_{\text{tot}}(\bar{S})} \right) .$$

*Proof.* First, observe that $d_{\text{tot}}(S) = \sum_{t=1}^{T} |m_t \cap S|$ for any $S \subseteq [T]$. Also note that the bound trivially holds if $\sigma_{\max} = 0$; hence, assume $\sigma_{\max} \geq 1$ without loss of generality. Let $t^*$ be any round such that $|m_{t^*}| = \sigma_{\max}$. Consider any $S \subseteq [T]$, and define $A = m_{t^*} \cap S$ and $B = m_{t^*} \cap \bar{S}$. If $|A| \geq (\sqrt{2} - 1)|m_{t^*}|$, then

$$|S| + \sqrt{d_{\text{tot}}(\bar{S})} \geq |S| \geq |A| \geq (\sqrt{2} - 1)\sigma_{\max} .$$

Otherwise, we have that $|B| > (2 - \sqrt{2})|m_{t^*}|$. Hence, denote $B = \{t_1, \ldots, t_{|B|}\}$ such that $t_1 < \cdots < t_{|B|}$ and observe that $|m_{t_i+1} \cap B| \geq i$ for any $t_i \in B$. We can consequently prove that

$$|S| + \sqrt{d_{\text{tot}}(\bar{S})} \geq \sqrt{d_{\text{tot}}(\bar{S})} = \sqrt{\sum_{t=1}^{T} |m_t \cap \bar{S}|} \geq \sqrt{\sum_{t \in B} |m_{t+1} \cap B|} \geq \sqrt{\sum_{i=1}^{|B|} i} \geq \frac{|B|}{\sqrt{2}} > (\sqrt{2} - 1)\sigma_{\max} ,$$

which concludes the proof as $\frac{1}{\sqrt{2}-1} \leq 2\sqrt{2}$. $\square$

**Lemma A.8.** *Let $\sigma_{\max}^S = \max_{\tau \in [T]} |m_\tau \cap S|$ and $\bar{S} = [T] \setminus S$ for any $S \subseteq [T]$. Then,*

$$\sigma_{\max} = \min_{S \subseteq [T]} \left( |S| + \sigma_{\max}^{\bar{S}} \right) .$$

*Proof.* First, it trivially holds that

$$\sigma_{\max} \geq \min_{S \subseteq [T]} \left( |S| + \sigma_{\max}^{\bar{S}} \right) .$$

We now only need to show the inequality in the other direction. Consider any $S \subseteq [T]$ and let $t^*$ be any round such that $|m_{t^*}| = \sigma_{\max}$. Then,

$$|S| + \sigma_{\max}^{\bar{S}} \geq |S| + |m_{t^*} \cap \bar{S}| = |S| + |m_{t^*} \setminus S| \geq |m_{t^*}| = \sigma_{\max} ,$$

which concludes the proof. $\square$

The following lemma further illustrates the relationship between $\sigma_{\max}$ and $\sqrt{d_{\text{tot}}}$ in a more concrete way.

**Lemma A.9.** *There exists a delay sequence $(d_t)_{t \in [T]}$ such that $\sigma_{\max} \geq \sqrt{1.5 \cdot d_{\text{tot}}}$. In addition, there also exists a delay sequence such that $\sigma_{\max} = 1$ and $\sqrt{d_{\text{tot}}} = \sqrt{T}$.*

*Proof.* Given a positive integer $N > 5$, consider the sequence $(d_t)_{t \in [T]}$, where $d_t = N - t$ for all $t \leq N$ and $d_t = 0$ for all $t > N$. In this case, $\sigma_{\max} = \sigma_{N-1} = N - 1$ and $\sqrt{1.5 \cdot d_{\text{tot}}} = \sqrt{\frac{3N(N-1)}{4}} \leq N - 1$. On the other hand, consider the sequence where $d_t = 1$ for all $t \in [T]$. In this case, $\sigma_{\max} = 1$ and $\sqrt{d_{\text{tot}}} = \sqrt{T}$. $\square$

On a similar note, we show another similar result depicting the relationship between $d_{\max}$ and $\sqrt{d_{\text{tot}}}$.

**Lemma A.10.** *There exists a delay sequence $(d_t)_{t \in [T]}$ such that $d_{\max} = T$ and $\sqrt{d_{\text{tot}}} = \sqrt{T}$. In addition, there also exists a delay sequence such that $d_{\max} = 1$ and $\sqrt{d_{\text{tot}}} = \sqrt{T}$.*

*Proof.* Consider the sequence $(d_t)_{t \in [T]}$ where one round $t_0 \leq T/2$ with $d_{t_0} = T - t_0$ and all the other rounds $d_t = 0$ for $t \neq t_0$, then we can choose $t_0 = 1$ and have $d_{\max} = T$ and $\sqrt{d_{\text{tot}}} = \sqrt{T}$. On the other hand, consider the sequence where $d_t = 1$ for all $t \in [T]$, then $d_{\max} = 1$ and $\sqrt{d_{\text{tot}}} = \sqrt{T}$. $\square$

## B. Omitted details in Section 3

In this section, we show the omitted details in Section 3. For completeness, we restate the theorem and provide its proof.

**Theorem 3.1.** *Assume that $f_1, \ldots, f_T$ are $\lambda$-strongly convex with respect to the Euclidean norm $\|\cdot\|_2$. Then, under Assumption 2.3, Algorithm 1 guarantees that*

$$\mathrm{Reg}_T = \mathcal{O}\left(\frac{G^2}{\lambda}\left(\ln T + \min\left\{\sigma_{\max} \ln T, \sqrt{d_{\mathrm{tot}}}\right\}\right)\right).$$

*Proof.* First of all, define

$$F_t(x) = \sum_{\tau \in o_t} \langle g_\tau, x \rangle + \frac{\lambda}{2}\sum_{\tau=1}^{t-1}\|x - x_\tau\|_2^2 \qquad \text{and} \qquad F_t^\star(x) = \sum_{\tau=1}^{t-1}\left(\langle g_\tau, x \rangle + \frac{\lambda}{2}\|x - x_\tau\|_2^2\right)$$

for any $t \geq 1$. Observe that $x_t \in \arg\min_{x \in \mathcal{X}} F_t(x)$ and additionally define $x_t^\star \in \arg\min_{x \in \mathcal{X}} F_t^\star(x)$ for $t \geq 2$, while $x_1^\star = x_1$ (since $F_1^\star(x) = F_1(x)$). The sequence $(x_t^\star)_{t \geq 1}$ represents the "cheating" sequence that uses the gradients from all rounds up to $t - 1$, including those from rounds in $m_t$ that are yet to be received because of the delays. As mentioned in Section 3, we decompose the regret as follows:

$$\mathrm{Reg}_T(u) = \sum_{t=1}^{T}\big(f_t(x_t) - f_t(u)\big) \leq \sum_{t=1}^{T}\left(\langle g_t, x_t - u \rangle - \frac{\lambda}{2}\|x_t - u\|_2^2\right)$$

$$= \underbrace{\sum_{t=1}^{T}\langle g_t, x_t^\star - u\rangle}_{\mathrm{Reg}_T^\star(u)} + \underbrace{\sum_{t=1}^{T}\langle g_t, x_t - x_t^\star\rangle}_{\mathtt{Drift}_T} - \frac{\lambda}{2}\sum_{t=1}^{T}\|x_t - u\|_2^2, \tag{14}$$

where the first inequality follows from the $\lambda$-strong convexity of $f_t$. Next, we analyze the cheating term $\mathrm{Reg}_T^\star(u)$ and the drift term $\mathtt{Drift}_T$ individually, and their respective upper bounds will then be combined to derive the final regret bound.

To analyze $\mathrm{Reg}_T^\star(u)$, first define $\psi_t(x) = \frac{\lambda}{2}\sum_{\tau=1}^{t-1}\|x - x_\tau\|_2^2$ for $t \geq 1$. We can therefore rewrite both $F_t(x) = \sum_{\tau \in o_t}\langle g_\tau, x\rangle + \psi_t(x)$ and $F_t^\star(x) = \sum_{\tau=1}^{t-1}\langle g_\tau, x\rangle + \psi_t(x)$. Hence, applying Lemma A.1, we can bound $\mathrm{Reg}_T^\star(u)$ as follows:

$$\mathrm{Reg}_T^\star(u) = \sum_{t=1}^{T}\langle g_t, x_t^\star - u\rangle$$

$$= \psi_{T+1}(u) - \min_{x \in \mathcal{X}}\psi_1(x) + \sum_{t=1}^{T}\left[F_t^\star(x_t^\star) - F_{t+1}^\star(x_{t+1}^\star) + \langle g_t, x_t^\star\rangle\right] + F_{T+1}^\star(x_{T+1}^\star) - F_{T+1}^\star(u)$$

$$\leq \psi_{T+1}(u) + \sum_{t=1}^{T}\left[\big(F_t^\star(x_t^\star) + \langle g_t, x_t^\star\rangle\big) - \big(F_t^\star(x_{t+1}^\star) + \langle g_t, x_{t+1}^\star\rangle\big) - \psi_{t+1}(x_{t+1}^\star) + \psi_t(x_{t+1}^\star)\right], \tag{15}$$

where the last inequality holds because $F_{T+1}^\star(x_{T+1}^\star) \leq F_{T+1}^\star(u)$ by optimality of $x_{T+1}^\star$, together with the non-negativity of $\psi_1$.

Focus on the difference between the terms $F_t^\star(x_t^\star) + \langle g_t, x_t^\star\rangle$ and $F_t^\star(x_{t+1}^\star) + \langle g_t, x_{t+1}^\star\rangle$ within the sum in the right-hand side of Equation (15). Applying Lemma A.2 for $z_1 = x_{t+1}^\star$ with $A_1 = \frac{\lambda t}{2}I$ and $w_1 = \sum_{\tau \leq t} g_\tau$, and $z_2 = x_t^\star$ with $A_2 = \frac{\lambda(t-1)}{2}I$ and $w_2 = \sum_{\tau \leq t-1} g_\tau$, we have that

$$(2t-1)\frac{\lambda}{2}\|x_t^\star - x_{t+1}^\star\|_2^2 = \|x_t^\star - x_{t+1}^\star\|_{A_1}^2 + \|x_t^\star - x_{t+1}^\star\|_{A_2}^2$$

$$\leq \langle g_t, x_t^\star - x_{t+1}^\star\rangle + \frac{\lambda}{2}\|x_t^\star - x_t\|_2^2 - \frac{\lambda}{2}\|x_{t+1}^\star - x_t\|_2^2$$

$$\leq \|g_t\|_2\|x_t^\star - x_{t+1}^\star\|_2 + \frac{\lambda}{2}\|x_t^\star - x_t\|_2^2,$$

where we used the Cauchy-Schwarz inequality in the last step. By straightforward calculations, we can show that the above inequality implies that

$$\|x_t^\star - x_{t+1}^\star\|_2 \le \frac{2\|g_t\|_2}{\lambda(2t-1)} + \frac{\|x_t^\star - x_t\|_2}{\sqrt{2t-1}} \le \frac{2\|g_t\|_2}{\lambda(2t-1)} + \|x_t^\star - x_t\|_2 \ . \tag{16}$$

We can leverage this inequality to show that

$$
\begin{aligned}
\left(F_t^\star(x_t^\star) + \langle g_t, x_t^\star \rangle\right) - \left(F_t^\star(x_{t+1}^\star) + \langle g_t, x_{t+1}^\star \rangle\right) &\le \langle g_t, x_t^\star - x_{t+1}^\star \rangle && (F_t^\star(x_t^\star) \le F_t^\star(x_{t+1}^\star)) \\
&\le \|g_t\|_2 \|x_t^\star - x_{t+1}^\star\|_2 && \text{(Cauchy-Schwarz)} \\
&\le \frac{2\|g_t\|_2^2}{\lambda(2t-1)} + \|g_t\|_2\|x_t^\star - x_t\|_2 \ , && \text{(Equation (16))}
\end{aligned}
$$

where the first inequality is due to the optimality of $x_t^\star$ with respect to $F_t^\star$. Plugging the above into the bound on $\mathrm{Reg}_T^\star(u)$ from Equation (15), we obtain

$$
\begin{aligned}
\mathrm{Reg}_T^\star(u) &\le \psi_{T+1}(u) + \sum_{t=1}^T \left[ \frac{2\|g_t\|_2^2}{\lambda(2t-1)} + \|g_t\|_2\|x_t^\star - x_t\|_2 + \psi_t(x_{t+1}^\star) - \psi_{t+1}(x_{t+1}^\star) \right] \\
&= \frac{\lambda}{2}\sum_{t=1}^T \|x_t - u\|_2^2 + \sum_{t=1}^T \left[ \frac{2\|g_t\|_2^2}{\lambda(2t-1)} + \|g_t\|_2\|x_t^\star - x_t\|_2 - \frac{\lambda}{2}\|x_{t+1}^\star - x_t\|_2^2 \right] \\
&\le \frac{\lambda}{2}\sum_{t=1}^T \|x_t - u\|_2^2 + \frac{G^2}{\lambda}\sum_{t=1}^T \frac{2}{2t-1} + G\sum_{t=1}^T \|x_t^\star - x_t\|_2 \\
&\le \frac{\lambda}{2}\sum_{t=1}^T \|x_t - u\|_2^2 + \frac{G^2}{\lambda}\ln(2T+1) + G\sum_{t=1}^T \|x_t^\star - x_t\|_2 \ , \tag{17}
\end{aligned}
$$

where the equality is due to the definition of $\psi_t$, while the second inequality follows from $\|g_t\|_2 \le G$ by Assumption 2.3.

Observe that, given such a bound on the cheating term, we now have to consider three different terms as shown in Equation (17). While the second one is a desirable logarithmic term, and the first one is negligible since it will be canceled when plugging this bound on $\mathrm{Reg}_T^\star(u)$ into Equation (14), the third one needs some further analysis. Interestingly enough, this latter term involves a difference between $x_t^\star$ and $x_t$, in an analogous way as in the drift term $\mathtt{Drift}_T$. We indeed show that we can handle both terms in the same way.

We thus move to the analysis of the $\mathtt{Drift}_T$ term. One can immediately observe that, by Cauchy-Schwarz and by Assumption 2.3,

$$\mathtt{Drift}_T = \sum_{t=1}^T \langle g_t, x_t - x_t^\star \rangle \le \sum_{t=1}^T \|g_t\|_2\|x_t^\star - x_t\|_2 \le G\sum_{t=1}^T \|x_t^\star - x_t\|_2 \ . \tag{18}$$

While it immediately follows that $\|x_1^\star - x_1\|_2 = 0$ by definition of $x_1^\star$, we require some additional effort when studying the other norms $\|x_t^\star - x_t\|_2$ for $t \ge 2$. To this end, we rely once more on Lemma A.2 for $z_1 = x_t^\star$ with $w_1 = \sum_{\tau \le t-1} g_\tau$ and $z_2 = x_t$ with $w_2 = \sum_{\tau \in o_t} g_\tau$, using $A = (t-1)\frac{\lambda}{2}I$, and show that

$$\frac{\lambda(t-1)}{2}\|x_t^\star - x_t\|_2^2 = \|x_t^\star - x_t\|_A^2 \le \frac{1}{4}\left\|\sum_{\tau \in m_t} g_\tau\right\|_{A^{-1}}^2 = \frac{1}{2\lambda(t-1)}\left\|\sum_{\tau \in m_t} g_\tau\right\|_2^2 \ .$$

We can thus rewrite this inequality in the following way:

$$\|x_t^\star - x_t\|_2 \le \frac{1}{\lambda(t-1)}\left\|\sum_{\tau \in m_t} g_\tau\right\|_2 \le \frac{1}{\lambda(t-1)}\sum_{\tau \in m_t} \|g_\tau\|_2 \le \frac{G|m_t|}{\lambda(t-1)} \ , \tag{19}$$

where we used once again that $\|g_\tau\|_2 \le G$ by Assumption 2.3. The above considerations consequently imply that the sum of interest for bounding $\mathtt{Drift}_T$ satisfies

$$\sum_{t=1}^T \|x_t^\star - x_t\|_2 \le \frac{G}{\lambda}\sum_{t=2}^T \frac{|m_t|}{t-1} \ . \tag{20}$$

The sum on the right-hand side of the above inequality can be immediately bounded as

$$\sum_{t=2}^{T} \frac{|m_t|}{t-1} \leq \sigma_{\max} \sum_{t=2}^{T} \frac{1}{t-1} \leq \sigma_{\max} \ln(2T) \tag{21}$$

by definition of $\sigma_{\max}$. Furthermore, by using the fact that $\sum_{\tau \leq t} |m_\tau| \leq (t-1)^2$ since $m_\tau \subseteq [\tau-1]$ for any $\tau$, we can prove at the same time that

$$\sum_{t=2}^{T} \frac{|m_t|}{t-1} = \sum_{t=2}^{T} \frac{|m_t|}{\sqrt{(t-1)^2}} \leq \sum_{t=2}^{T} \frac{|m_t|}{\sqrt{\sum_{\tau \leq t} |m_\tau|}} \leq 2\sqrt{\sum_{t=1}^{T} |m_t|} \leq 2\sqrt{d_{\text{tot}}} , \tag{22}$$

where the second inequality is due to Orabona (2025, Lemma 4.13).

Combining all the results gathered so far, we can finally derive the overall regret bound as follows. In particular, for any $u \in \mathcal{X}$, we have

$$\begin{aligned}
\text{Reg}_T(u) &\leq \text{Reg}_T^*(u) + \texttt{Drift}_T - \frac{\lambda}{2} \sum_{t=1}^{T} \|x_t - u\|_2^2 &\text{(Equation (14))} \\
&\leq \frac{G^2}{\lambda} \ln(2T+1) + G \sum_{t=1}^{T} \|x_t^\star - x_t\|_2 + \texttt{Drift}_T &\text{(Equation (17))} \\
&\leq \frac{G^2}{\lambda} \ln(2T+1) + 2G \sum_{t=1}^{T} \|x_t^\star - x_t\|_2 &\text{(Equation (18))} \\
&\leq \frac{G^2}{\lambda} \ln(2T+1) + \frac{2G^2}{\lambda} \sum_{t=2}^{T} \frac{|m_t|}{t-1} &\text{(Equation (20))} \\
&\leq \frac{G^2}{\lambda} \ln(2T+1) + \frac{2G^2}{\lambda} \min\left\{\sigma_{\max} \ln(2T), 2\sqrt{d_{\text{tot}}}\right\} &\text{(Equations (21) and (22))} \\
&= \mathcal{O}\left(\frac{G^2}{\lambda}\left(\ln T + \min\left\{\sigma_{\max} \ln T, \sqrt{d_{\text{tot}}}\right\}\right)\right) . &\square
\end{aligned}$$

## C. Omitted details from Section 4

In this section, we show the omitted details from Section 4. To do so, we first introduce the following useful lemma that will be crucial in the regret analysis of Algorithm 2. It essentially corresponds to the standard elliptical potential lemma, but here adapted to the presence of delays.

**Lemma C.1.** *Let $\phi > 0$, $L > 0$, and $0 < \eta_0 \leq \eta_1 \leq \cdots \leq \eta_N$. For any $t \in [N]$, let $a_t \in \mathbb{R}^n$ such that $\|a_t\|_2 \leq L$ and define $A_t = \eta_t I + \phi \sum_{\tau \leq t} a_\tau a_\tau^\top$. Then, it holds that*

$$\sum_{t=1}^{N} \|a_t\|_{A_{t-1}^{-1}} \left(\sum_{\tau \in m_t} \|a_\tau\|_{A_{t-1}^{-1}}\right) \leq \frac{2n d_{\max}^{\leq N}}{\phi} \left(\frac{\phi L^2}{\eta_0} + 1\right) \ln\left(1 + \frac{\phi L^2 N}{\eta_0 n}\right) ,$$

*and that*

$$\sum_{t=1}^{N} \|a_t\|_{A_t^{-1}} \left(\sum_{\tau \in m_t} \|a_\tau\|_{A_t^{-1}}\right) \leq \frac{2n d_{\max}^{\leq N}}{\phi} \ln\left(1 + \frac{\phi L^2 N}{\eta_0 n}\right) .$$

*Proof.* Define $B_t = \frac{1}{\phi} A_t$ and $C_t = B_t - \frac{\eta_t - \eta_0}{\phi} I \preceq B_t$ for any $t \in [N]$. By the AM-GM inequality, we first show that

$$\sum_{t=1}^{N} \|a_t\|_{A_{t-1}^{-1}} \sum_{\tau \in m_t} \|a_\tau\|_{A_{t-1}^{-1}} \leq \sum_{t=1}^{N} \left(\frac{|m_t|}{2} \|a_t\|_{A_{t-1}^{-1}}^2 + \frac{1}{2} \sum_{\tau \in m_t} \|a_\tau\|_{A_{t-1}^{-1}}^2\right)$$

$$\leq \sum_{t=1}^{N} \left( \frac{|m_t|}{2} \|a_t\|_{A_{t-1}^{-1}}^2 + \frac{1}{2} \sum_{\tau \in m_t} \|a_\tau\|_{A_{\tau-1}^{-1}}^2 \right)$$

$$= \frac{1}{\phi} \sum_{t=1}^{N} \left( \frac{|m_t|}{2} \|a_t\|_{B_{t-1}^{-1}}^2 + \frac{1}{2} \sum_{\tau \in m_t} \|a_\tau\|_{B_{\tau-1}^{-1}}^2 \right),$$

where we also used the fact that $A_{\tau-1} \preceq A_{t-1}$ for any $\tau < t$. Now observe that

$$\sum_{t=1}^{N} |m_t| \cdot \|a_t\|_{B_{t-1}^{-1}}^2 \leq d_{\max}^{\leq N} \sum_{t=1}^{N} \|a_t\|_{B_{t-1}^{-1}}^2$$

since $|m_t| \leq d_{\max}^{\leq N}$ for $t \leq N$. Similarly, we can show that

$$\sum_{t=1}^{N} \sum_{\tau \in m_t} \|a_\tau\|_{B_{\tau-1}^{-1}}^2 = \sum_{t=1}^{N} d_t \|a_t\|_{B_{t-1}^{-1}}^2 \leq d_{\max}^{\leq N} \sum_{t=1}^{N} \|a_t\|_{B_{t-1}^{-1}}^2$$

as for any $\tau \in [N]$ there are no more than $d_\tau$ rounds $t$ such that $\tau \in m_t$. Putting these results together, we obtain that

$$\sum_{t=1}^{N} \left( \frac{|m_t|}{2} \|a_t\|_{B_{t-1}^{-1}}^2 + \frac{1}{2} \sum_{\tau \in m_t} \|a_\tau\|_{B_{\tau-1}^{-1}}^2 \right) \leq d_{\max}^{\leq N} \sum_{t=1}^{N} \|a_t\|_{B_{t-1}^{-1}}^2 \leq d_{\max}^{\leq N} \sum_{t=1}^{N} \|a_t\|_{C_{t-1}^{-1}}^2 .$$

By the fact that $\|a_t\|_{C_{t-1}^{-1}}^2 \leq \frac{\phi L^2}{\eta_0}$, we can use Lemma 19.4 in Lattimore & Szepesvári (2020) and show that

$$\sum_{t=1}^{N} \|a_t\|_{C_{t-1}^{-1}}^2 \leq \left( \frac{\phi L^2}{\eta_0} + 1 \right) \sum_{t=1}^{N} \min\left\{ 1, \|a_t\|_{C_{t-1}^{-1}}^2 \right\} \leq 2n \left( \frac{\phi L^2}{\eta_0} + 1 \right) \ln\left( 1 + \frac{L^2 N}{\eta_0 n} \right) .$$

Concatenating all the above results concludes the proof of the first inequality.

For the second inequality, similar steps suffice to prove it, but with a different observation that now $\|a_t\|_{C_t^{-1}}^2 \leq \min\left\{ 1, \|a_t\|_{C_{t-1}^{-1}}^2 \right\}$ because

$$\|a_t\|_{C_t^{-1}}^2 \leq a_t^\top \left( \upsilon I + a_t a_t^\top \right)^{-1} a_t = a_t^\top \left( \frac{1}{\upsilon} I - \frac{a_t a_t^\top}{\upsilon^2 + \upsilon \|a_t\|_2^2} \right) a_t = \frac{\|a_t\|_2^2}{\upsilon} - \frac{\|a_t\|_2^4}{\upsilon^2 + \upsilon \|a_t\|_2^2} = \frac{\|a_t\|_2^2}{\upsilon + \|a_t\|_2^2} \leq 1 ,$$

where we used the Sherman-Morrison formula in the first equality with $\upsilon = \eta_0/\phi$, and since $\|a_t\|_{C_t^{-1}} \leq \|a_t\|_{C_{t-1}^{-1}}$ given that $C_{t-1} \preceq C_t$. $\qquad\square$

For completeness, we restate Theorem 4.1, the main result of Section 4.1, and provide its proof.

**Theorem 4.1.** *Assume that $f_1, \ldots, f_T$ are $\alpha$-exp-concave and let $\beta = \frac{1}{2} \min\{\frac{1}{4GD}, \alpha\}$. Then, under Assumptions 2.3 and 2.4, Algorithm 2 with $0 < \eta_0 \leq \eta_1 \leq \cdots \leq \eta_T$ guarantees that*

$$\mathrm{Reg}_T = \mathcal{O}\left( \frac{n}{\beta} \ln\left( 1 + \frac{\beta G^2 T}{\eta_0 n} \right) + \eta_T D^2 + \min\{B_1, B_2\} \right),$$

*where $B_1 = \left( \frac{G^2}{\eta_0} + \frac{1}{\beta} \right) n d_{\max} \ln\left( 1 + \frac{\beta G^2 T}{\eta_0 n} \right)$ and $B_2 = G^2 \sum_{t=1}^{T} \frac{|m_t|}{\eta_{t-1}}$.*

*Proof.* First, in a similar way as in the proof of Theorem 3.1, we define

$$F_t(x) = \sum_{\tau \in o_t} \langle g_\tau, x \rangle + \psi_t(x) \qquad \text{and} \qquad F_t^\star(x) = \sum_{\tau=1}^{t-1} \langle g_\tau, x \rangle + \psi_t^\star(x),$$

where $\psi_t(x) = \frac{\eta_{t-1}}{2}\|x\|_2^2 + \frac{\beta}{2}\sum_{\tau \in O_t}\langle g_\tau, x - x_\tau\rangle^2$ and $\psi_t^\star(x) = \frac{\eta_{t-1}}{2}\|x\|_2^2 + \frac{\beta}{2}\sum_{\tau=1}^{t-1}\langle g_\tau, x - x_\tau\rangle^2$. Observe that $x_t \in \arg\min_{x \in \mathcal{X}} F_t(x)$, and define $x_t^\star \in \arg\min_{x \in \mathcal{X}} F_t^\star(x)$ for $t \geq 1$ to be the predictions following a similar update rule while using all the information up to round $t-1$. Similarly to the regret decomposition for the strongly convex case shown in Appendix B, we decompose the regret as follows:

$$\text{Reg}_T(u) = \sum_{t=1}^T (f_t(x_t) - f_t(u)) \leq \sum_{t=1}^T \left( \langle g_t, x_t - u\rangle - \frac{\beta}{2}\langle x_t - u, g_t\rangle^2 \right)$$

$$= \underbrace{\sum_{t=1}^T \langle g_t, x_t^\star - u\rangle}_{\text{Reg}_T^\star(u)} + \underbrace{\sum_{t=1}^T \langle g_t, x_t - x_t^\star\rangle}_{\text{Drift}_T} - \frac{\beta}{2}\sum_{t=1}^T \langle x_t - u, g_t\rangle^2 , \quad (23)$$

where the inequality holds thanks to Lemma A.3.

Let us begin the analysis of the "linearized" regret by first focusing on the cheating term $\text{Reg}_T^\star(u)$. Let $F_t'(x) = F_t^\star(x) + \langle g_t, x\rangle$ and define $x_t' \in \arg\min_{x \in \mathcal{X}} F_t'(x)$. Leveraging Lemma A.1 with $\ell_t(\cdot) = \langle g_t, \cdot\rangle$, we show that

$$\text{Reg}_T^\star(u) = \sum_{t=1}^T \langle g_t, x_t^\star - u\rangle$$

$$= \psi_{T+1}^\star(u) - \min_{x \in \mathcal{X}} \psi_1^\star(x) + \sum_{t=1}^T \left[ F_t^\star(x_t^\star) - F_{t+1}^\star(x_{t+1}^\star) + \langle g_t, x_t^\star\rangle \right] + F_{T+1}^\star(x_{T+1}^\star) - F_{T+1}^\star(u)$$

$$\leq \psi_{T+1}^\star(u) + \sum_{t=1}^T \left[ \left(F_t^\star(x_t^\star) + \langle g_t, x_t^\star\rangle\right) - \left(F_t^\star(x_{t+1}^\star) + \langle g_t, x_{t+1}^\star\rangle\right) - \psi_{t+1}^\star(x_{t+1}^\star) + \psi_t^\star(x_{t+1}^\star) \right]$$

$$\leq \psi_{T+1}^\star(u) + \sum_{t=1}^T \left[ F_t'(x_t^\star) - F_t'(x_t') + \psi_t^\star(x_{t+1}^\star) - \psi_{t+1}^\star(x_{t+1}^\star) \right] \quad \text{(definition of } F_t' \text{ and } x_t')$$

$$\leq \psi_{T+1}^\star(u) + \sum_{t=1}^T \left( F_t'(x_t^\star) - F_t'(x_t') \right), \quad (24)$$

where in the first inequality we used the facts that $F_{T+1}^\star(x_{T+1}^\star) \leq F_{T+1}^\star(u)$ and that $\psi_1^\star$ is nonnegative, while the last inequality is due to $\psi_t^\star(x_{t+1}^\star) \leq \psi_{t+1}^\star(x_{t+1}^\star)$. Applying now Lemma A.2, we have $\|x_t^\star - x_t'\|_{A_{t-1}} \leq \|g_t\|_{A_{t-1}^{-1}}$, where $A_{t-1} = \eta_{t-1}I + \beta\sum_{\tau=1}^{t-1} g_\tau g_\tau^\top$. This further means that

$$\begin{aligned}
F_t'(x_t^\star) - F_t'(x_t') &\leq \langle \nabla F_t'(x_t^\star), x_t^\star - x_t'\rangle && \text{(convexity of } F_t') \\
&= \langle \nabla F_t^\star(x_t^\star) + g_t, x_t^\star - x_t'\rangle && \text{(definition of } F_t') \\
&\leq \langle g_t, x_t^\star - x_t'\rangle && \text{(first-order optimality)} \\
&\leq \min\left\{ \|g_t\|_2\|x_t^\star - x_t'\|_2, \|g_t\|_{A_{t-1}^{-1}}\|x_t^\star - x_t'\|_{A_{t-1}} \right\} && \text{(Cauchy-Schwarz inequality)} \\
&\leq \min\left\{ GD, \|g_t\|_{A_{t-1}^{-1}}\|x_t^\star - x_t'\|_{A_{t-1}} \right\} && \text{(Assumptions 2.3 and 2.4)} \\
&\leq \min\left\{ GD, \|g_t\|_{A_{t-1}^{-1}}^2 \right\} . && (25)
\end{aligned}$$

We now focus on the sum of terms on the right-hand side of Equation (24). Because $\eta_t$ is non-decreasing by assumption, we have

$$\sum_{t=1}^T \left( F_t'(x_t^\star) - F_t'(x_t') \right) \leq \sum_{t=1}^T \min\left\{ GD, \|g_t\|_{A_{t-1}^{-1}}^2 \right\} \quad \text{(Equation (25))}$$

$$\leq \sum_{t=1}^T \min\left\{ GD, \frac{1}{\beta}\|g_t\|_{(\frac{\eta_0}{\beta}I + \sum_{\tau < t} g_\tau g_\tau^\top)^{-1}}^2 \right\} \quad (\eta_{t-1}I \succeq \eta_0 I)$$

$$\leq \max\left\{GD, \frac{1}{\beta}\right\} \sum_{t=1}^{T} \min\left\{1, \|g_t\|^2_{(\frac{\eta_0}{\beta} I + \sum_{\tau<t} g_\tau g_\tau^\top)^{-1}}\right\}$$

$$\leq \left(GD + \frac{1}{\beta}\right) n \ln\left(1 + \frac{\beta G^2 T}{n\eta_0}\right) , \tag{26}$$

where the last inequality follows by [Lattimore & Szepesvári (2020)](#), Lemma 19.4). Combining the previous inequalities, we can show that $\mathrm{Reg}^\star_T(u)$ satisfies

$$\mathrm{Reg}^\star_T(u) \leq \psi^\star_{T+1}(u) + \sum_{t=1}^{T}\left(F'_t(x^\star_t) - F'_t(x'_t)\right) \qquad \text{(Equation (24))}$$

$$\leq \psi^\star_{T+1}(u) + \frac{\beta}{2}\sum_{t=1}^{T}(\langle g_t, u - x_t\rangle)^2 + \left(GD + \frac{1}{\beta}\right) n \ln\left(1 + \frac{\beta G^2 T}{n\eta_0}\right) \qquad \text{(Equation (26))}$$

$$= \frac{\eta_T}{2}\|u\|_2^2 + \frac{\beta}{2}\sum_{t=1}^{T}(\langle g_t, u - x_t\rangle)^2 + \left(GD + \frac{1}{\beta}\right) n \ln\left(1 + \frac{\beta G^2 T}{n\eta_0}\right) , \tag{27}$$

where we simply replace $\psi^\star_{T+1}$ with its definition in the last step.

We thus move to the analysis of the $\mathtt{Drift}_T$ term. Using the Cauchy-Schwarz inequality, we have

$$\mathtt{Drift}_T = \sum_{t=1}^{T} \langle g_t, x_t - x^\star_t\rangle \leq \sum_{t=1}^{T} \|g_t\|_{A^{-1}_{t-1}} \cdot \|x_t - x^\star_t\|_{A_{t-1}} . \tag{28}$$

Applying [Lemma A.2](#), we obtain that

$$F^\star_t(x_t) - F^\star_t(x^\star_t) \geq \frac{1}{2}\|x_t - x^\star_t\|^2_{A_{t-1}} \qquad \text{and} \qquad F_t(x^\star_t) - F_t(x_t) \geq \frac{1}{2}\|x_t - x^\star_t\|^2_{A_{o_t}} ,$$

where $A_{o_t} = \eta_{t-1} I + \beta \sum_{\tau \in o_t} g_\tau g_\tau^\top$. Summing the above inequalities, and replacing $F^\star_t$ and $F_t$ with their definitions, it follows that

$$\frac{1}{2}\|x_t - x^\star_t\|^2_{A_{o_t}} + \frac{1}{2}\|x_t - x^\star_t\|^2_{A_{t-1}}$$

$$\leq \sum_{\tau=1}^{t-1}\langle g_\tau, x_t\rangle - \sum_{\tau \in o_t}\langle g_\tau, x^\star_t\rangle + \sum_{\tau=1}^{t-1}\langle g_\tau, x^\star_t\rangle - \sum_{\tau \in o_t}\langle g_\tau, x_t\rangle$$

$$+ \frac{\beta}{2}\left(\sum_{\tau=1}^{t-1}\langle g_\tau, x_t - x_\tau\rangle^2 - \sum_{\tau=1}^{t-1}\langle g_\tau, x^\star_t - x_\tau\rangle^2 + \sum_{\tau \in o_t}\langle g_\tau, x^\star_t - x_\tau\rangle^2 - \sum_{\tau \in o_t}\langle g_\tau, x_t - x_\tau\rangle^2\right)$$

$$\leq \sum_{\tau \in m_t}\langle g_\tau, x_t - x^\star_t\rangle + \frac{\beta}{2}\left(\sum_{\tau \in m_t}\langle g_\tau, x_t - x_\tau\rangle^2 - \sum_{\tau \in m_t}\langle g_\tau, x^\star_t - x_\tau\rangle^2\right)$$

$$= \sum_{\tau \in m_t}\langle g_\tau, x_t - x^\star_t\rangle + \frac{\beta}{2}\left(\sum_{\tau \in m_t}\langle g_\tau, x_t - x^\star_t\rangle \cdot \langle g_\tau, x_t + x^\star_t - 2x_\tau\rangle\right)$$

$$\leq \sum_{\tau \in m_t}|\langle g_\tau, x_t - x^\star_t\rangle| + \frac{\beta}{2}\sum_{\tau \in m_t}|\langle g_\tau, x_t - x^\star_t\rangle||\langle g_\tau, x_t + x^\star_t - 2x_\tau\rangle|$$

$$\leq (1 + 2GD\beta)\sum_{\tau \in m_t}|\langle g_\tau, x_t - x^\star_t\rangle| \qquad \text{(Assumptions 2.3 and 2.4)}$$

$$\leq (1 + 2GD\beta)\left(\sum_{\tau \in m_t}\|g_\tau\|_{A^{-1}_{t-1}}\right)\|x_t - x^\star_t\|_{A_{t-1}} \qquad \text{(Cauchy-Schwarz inequality)}$$

$$\leq \frac{5}{4}\left(\sum_{\tau \in m_t}\|g_\tau\|_{A^{-1}_{t-1}}\right)\|x_t - x^\star_t\|_{A_{t-1}} \qquad (\beta \leq \frac{1}{8GD})$$

$$\leq 2 \left( \sum_{\tau \in m_t} \|g_\tau\|_{A_{t-1}^{-1}} \right) \|x_t - x_t^\star\|_{A_{t-1}} .$$

Rearranging the terms, we can obtain that $\|x_t - x_t^\star\|_{A_{t-1}} \leq 4 \sum_{\tau \in m_t} \|g_\tau\|_{A_{t-1}^{-1}}$. Plugging this inequality into $\texttt{Drift}_T$, we have

$$\texttt{Drift}_T \leq \sum_{t=1}^{T} \|g_t\|_{A_{t-1}^{-1}} \cdot \|x_t - x_t^\star\|_{A_{t-1}} \qquad \text{(Equation (28))}$$

$$\leq 4 \sum_{t=1}^{T} \|g_t\|_{A_{t-1}^{-1}} \left( \sum_{\tau \in m_t} \|g_\tau\|_{A_{t-1}^{-1}} \right)$$

$$\leq 8 d_{\max}^{\leq T} n \left( \frac{G^2}{\eta_0} + \frac{1}{\beta} \right) \ln \left( 1 + \frac{\beta T G^2}{n \eta_0} \right) , \qquad (29)$$

where the last inequality is due to Lemma C.1. On the other hand, we can also bound $\texttt{Drift}_T$ in a different way:

$$\texttt{Drift}_T \leq \sum_{t=1}^{T} \|g_t\|_{A_{t-1}^{-1}} \cdot \|x_t - x_t^\star\|_{A_{t-1}}$$

$$\leq 4 \sum_{t=1}^{T} \|g_t\|_{A_{t-1}^{-1}} \left( \sum_{\tau \in m_t} \|g_\tau\|_{A_{t-1}^{-1}} \right)$$

$$\leq 4 G^2 \sum_{t=1}^{T} \frac{|m_t|}{\eta_{t-1}} , \qquad (30)$$

where in the last step we use the fact that $\|g_s\|_{A_{t-1}^{-1}}^2 \leq \frac{G^2}{\eta_{t-1}}$ for any $s \in [T]$, also due to Assumption 2.3. Combining all bounds together, we finally obtain that

$$\text{Reg}_T(u) \leq \text{Reg}_T^\star(u) + \texttt{Drift}_T - \frac{\beta}{2} \sum_{t=1}^{T} \langle x_t - u, g_t \rangle^2 \qquad \text{(Equation (23))}$$

$$\leq \frac{\eta_T}{2} \|u\|_2^2 + \left( GD + \frac{1}{\beta} \right) n \ln \left( 1 + \frac{\beta G^2 T}{n \eta_0} \right) + \texttt{Drift}_T \qquad \text{(Equation (27))}$$

$$\leq \frac{\eta_T}{2} \|u\|_2^2 + \left( GD + \frac{1}{\beta} \right) n \ln \left( 1 + \frac{\beta G^2 T}{n \eta_0} \right)$$

$$+ 4 \min \left\{ 2 d_{\max}^{\leq T} n \left( \frac{G^2}{\eta_0} + \frac{1}{\beta} \right) \ln \left( 1 + \frac{\beta G^2 T}{\eta_0 n} \right), G^2 \sum_{t=1}^{T} \frac{|m_t|}{\eta_{t-1}} \right\} \qquad \text{(Equations (29) and (30))}$$

$$= \mathcal{O} \left( \frac{n}{\beta} \ln \left( 1 + \frac{\beta G^2 T}{\eta_0 n} \right) + \eta_T D^2 + \min \{B_1, B_2\} \right) , \qquad \text{(Assumption 2.4)}$$

where

$$B_1 = \left( \frac{G^2}{\eta_0} + \frac{1}{\beta} \right) n d_{\max} \ln \left( 1 + \frac{\beta G^2 T}{\eta_0 n} \right) \qquad \text{and} \qquad B_2 = G^2 \sum_{t=1}^{T} \frac{|m_t|}{\eta_{t-1}}$$

are defined as in the theorem statement, and we used the fact that $GD \leq \frac{1}{\beta}$. $\qquad \square$

The following corollary is a restatement of Corollary 4.2, which shows that via an adaptive tuning of the learning rate used by Algorithm 2, we are able to guarantee $\mathcal{O}(\min\{d_{\max} \ln T, \sqrt{d_{\text{tot}}}\})$ regret.

**Corollary 4.2.** *Assume that $f_1, \ldots, f_T$ are $\alpha$-exp-concave and let $\beta = \frac{1}{2} \min\{\frac{1}{4GD}, \alpha\}$. Then, under Assumptions 2.3 and 2.4, Algorithm 2 with the adaptive learning rate $\eta_t = \min\{a_t, b_t\} + 1$, where $a_t$ and $b_t$ are defined in Equations (5) and (6), guarantees that*

$$\text{Reg}_T = \mathcal{O} \left( \frac{n}{\beta} \ln \left( 1 + \frac{\beta G^2 T}{n} \right) + D^2 + \min \{C_1, C_2\} \right),$$

*where $C_1 = \left(\frac{D}{G} + 1\right) \left(G^2 + \frac{1}{\beta}\right) nd_{\max} \ln\left(1 + \frac{\beta G^2 T}{n}\right)$ and $C_2 = \left(G^2 + GD\right)\left(\sqrt{d_{\text{tot}}} + 1\right)$.*

*Proof.* The adaptive learning rate is given by $\eta_0 = 1$ and $\eta_t = \min\{a_t, b_t\} + 1$ for all $t \geq 1$, where we recall that

$$a_t = \frac{2}{GD}\left(G^2 + \frac{1}{\beta}\right) nd_{\max}^{\leq t} \ln\left(1 + \frac{\beta G^2 T}{n}\right) \qquad \text{and} \qquad b_t = \frac{G}{D}\sqrt{\sum_{s=1}^{t} |m_s| + |m_t| + 1}\,,$$

Note that $\eta_t$ is non-decreasing since $a_t$ and $b_t$ are non-decreasing. When $a_T \leq b_T$, we have

$$\mathrm{Reg}_T(u) \leq \left(GD + \frac{1}{\beta}\right) n \ln\left(1 + \frac{\beta GT}{n}\right) + D^2 + \left(\frac{2D}{G} + 8\right)\left(G^2 + \frac{1}{\beta}\right) nd_{\max} \ln\left(1 + \frac{\beta G^2 T}{n}\right), \tag{31}$$

where $\|u\|_2 \leq D$ by Assumption 2.4. When $a_T \geq b_T$, we instead have

$$\mathrm{Reg}_T(u) \leq \left(GD + \frac{1}{\beta}\right) n \ln\left(1 + \frac{\beta G^2 T}{n}\right) + D^2 + GD\left(\sqrt{\sum_{t=1}^{T} |m_t| + 1}\right)$$

$$+ \sum_{t=1}^{\tau^\star} \|g_t\|_{A_{t-1}^{-1}} \cdot \|x_t - x_t^\star\|_{A_{t-1}} + \sum_{t=\tau^\star+1}^{T} \|g_t\|_{A_{t-1}^{-1}} \cdot \|x_t - x_t^\star\|_{A_{t-1}},$$

where $\tau^\star$ is last round $a_{\tau^\star} \leq b_{\tau^\star}$. Hence, we have

$$\sum_{t=1}^{\tau^\star} \|g_t\|_{A_{t-1}^{-1}} \cdot \|x_t - x_t^\star\|_{A_{t-1}} \leq 8\left(G^2 + \frac{1}{\beta}\right) nd_{\max}^{\leq \tau^\star} \ln\left(1 + \frac{\beta G^2 T}{n}\right) \qquad \text{(Equation (29))}$$

$$\leq 8G^2\sqrt{\sum_{t=1}^{\tau^\star} |m_t| + |m_{\tau^\star}| + 1}$$

$$\leq 8G^2\left(\sqrt{\sum_{t=1}^{T} |m_t| + 1}\right) \tag{32}$$

Regarding the remaining rounds until $T$, we can also show that

$$\sum_{t=\tau^\star+1}^{T} \|g_t\|_{A_{t-1}^{-1}} \cdot \|x_t - x_t^\star\|_{A_{t-1}} \leq 4G^2 \sum_{t=\tau^\star+1}^{T} \frac{|m_t|}{\eta_{t-1}} \qquad \text{(Equation (30))}$$

$$\leq 4G^2 \sum_{t=\tau^\star+1}^{T} \frac{D|m_t|}{G\sqrt{\sum_{s=1}^{t-1} |m_s| + |m_{t-1}| + 1}}$$

$$\leq 8G^2 \sum_{t=\tau^\star+1}^{T} \frac{D|m_t|}{G\sqrt{\sum_{s=\tau^\star+1}^{t} |m_s|}}$$

$$\leq 8GD\sqrt{\sum_{t=\tau^\star+1}^{T} |m_t|}$$

$$\leq 8GD\sqrt{\sum_{t=1}^{T} |m_t|}, \tag{33}$$

where the last inequality is due to Orabona (2025, Lemma 4.13). Combining the above three inequalities together, we have

$$\mathrm{Reg}_T(u) \leq \left(GD + \frac{1}{\beta}\right) n \ln\left(1 + \frac{\beta G^2 T}{n}\right) + D^2 + \left(8G^2 + 9GD\right)\left(\sqrt{\sum_{t=1}^{T} |m_t| + 1}\right).$$

Finally, we obtain

$$\text{Reg}_T(u) \le \left(GD + \frac{1}{\beta}\right) n \ln\left(1 + \frac{\beta G^2 T}{n}\right) + D^2$$

$$+ \min\left\{\left(\frac{2D}{G} + 8\right)\left(G^2 d_{\max}^{\le T} + \frac{d_{\max}^{\le T}}{\beta}\right) n \ln\left(1 + \frac{\beta G^2 T}{n}\right), (8G^2 + 9GD)\left(\sqrt{d_{\text{tot}}} + 1\right)\right\}$$

$$= \mathcal{O}\left(\frac{1}{\beta} \ln\left(1 + \frac{\beta G^2 T}{n}\right) + D^2 + \min\{C_1, C_2\}\right),$$

where

$$C_1 = \left(\frac{D}{G} + 1\right)\left(G^2 + \frac{1}{\beta}\right) n d_{\max} \ln\left(1 + \frac{\beta G^2 T}{n}\right)$$

and

$$C_2 = (G^2 + GD)\left(\sqrt{d_{\text{tot}}} + 1\right)$$

as in the theorem statement. □

## D. Omitted details from Section 5

Here we present the omitted details from Section 5. For completeness, we restate the main result (Theorem 5.2) and provide its proof.

**Theorem 5.2.** *In the OLR problem with delayed labels under Assumption 5.1, Algorithm 3 guarantees for any $0 < \eta_0 \le \eta_1 \le \cdots \le \eta_T$ that*

$$\text{Reg}_T(u) \le \frac{\eta_T}{2}\|u\|_2^2 + nY^2 \ln\left(1 + \frac{Z^2 T}{\eta_0 n}\right)$$

$$+ \mathcal{O}\left(Y^2\left(\sigma_{\max} + \min\{M_1, M_2\}\right)\right),$$

*where $M_1 = n d_{\max} \ln\left(1 + \frac{Z^2 T}{\eta_0 n}\right)$ and $M_2 = Z^2 \sum_{t=1}^{T} \frac{|m_t|}{\eta_t}$.*

*Proof.* We begin by defining

$$F_t(x) = \sum_{\tau \in o_t} -y_\tau \langle z_\tau, x \rangle + \psi_t(x) \qquad \text{and} \qquad F_t^*(x) = \sum_{\tau=1}^{t-1} -y_\tau \langle z_\tau, x \rangle + \psi_t(x),$$

where $\psi_t(x) = \frac{1}{2}\sum_{\tau=1}^{t}\langle z_\tau, x \rangle^2 + \frac{\eta_t}{2}\|x\|_2^2$ for $t \in [T]$, and we let $\psi_{T+1} = \psi_T$. Observe that $x_t \in \arg\min_{x \in \mathbb{R}^n} F_t(x)$, and define $x_t^\star \in \arg\min_{x \in \mathbb{R}^n} F_t^\star(x)$ for $t \ge 1$ to be the predictions following a similar update rule while using all the information up to round $t-1$, including the labels $y_\tau$ for rounds $\tau \in m_t$ that the algorithm is missing because of the delays.

Similarly to the regret decomposition for the strongly convex case shown in Appendix B, we rewrite the regret as follows:

$$\text{Reg}_T(u) = \sum_{t=1}^{T}\left(f_t(\widetilde{x}_t) - f_t(u)\right) = \underbrace{\sum_{t=1}^{T}\left(f_t(x_t^\star) - f_t(u)\right)}_{\text{Reg}_T^\star(u)} + \underbrace{\sum_{t=1}^{T}\left(f_t(\widetilde{x}_t) - f_t(x_t^\star)\right)}_{\text{Drift}_T}, \tag{34}$$

where $\text{Reg}_T^\star(u)$ is the cheating regret for the iterates $x_1^\star, \ldots, x_T^\star$, while $\text{Drift}_T$ is a drift term that quantifies the influence of the missing labels on the regret because of the delayed feedback. Note that, contrarily to other regret analyses in this work, here $\text{Drift}_T$ is also affected by the clipping in the definition of $\widetilde{x}_t$.

Let us first analyze the cheating regret $\text{Reg}_T^\star(u)$. By the definition of the loss $f_t(x) = \frac{1}{2}\left(\langle z_t, x \rangle - y_t\right)^2$, we can rewrite the regret in the following way:

$$\text{Reg}_T^\star(u) = \sum_{t=1}^{T}\left(f_t(x_t^\star) - f_t(u)\right) = \frac{1}{2}\sum_{t=1}^{T}\langle z_t, x_t^\star \rangle^2 + \sum_{t=1}^{T}\left(-y_t\langle z_t, x_t^\star \rangle + y_t\langle z_t, u \rangle\right) - \frac{1}{2}\sum_{t=1}^{T}\langle z_t, u \rangle^2. \tag{35}$$

We can now move our focus on the central sum, which essentially corresponds to the regret of the same sequence $(x_t^\star)_{t \geq 1}$ against the comparator $u \in \mathbb{R}^n$, but with respect to the linear losses $x \mapsto -y_t \langle z_t, x \rangle$. Additionally define $F_t'(x) = F_t^\star(x) - y_t \langle z_t, x \rangle$ for notational convenience. Hence, we analyze the above-mentioned term by applying Lemma A.1, which yields

$$\sum_{t=1}^{T} \left( -y_t \langle z_t, x_t^\star \rangle + y_t \langle z_t, u \rangle \right)$$

$$= \psi_{T+1}(u) - \min_{x \in \mathbb{R}^n} \psi_1(x) + \sum_{t=1}^{T} \left[ F_t^\star(x_t^\star) - F_{t+1}^\star(x_{t+1}^\star) - y_t \langle z_t, x_{t+1}^\star \rangle \right] + F_{T+1}^\star(x_{T+1}^\star) - F_{T+1}^\star(u)$$

$$\leq \psi_{T+1}(u) + \sum_{t=1}^{T} \left[ F_t^\star(x_t^\star) - F_{t+1}^\star(x_{t+1}^\star) - y_t \langle z_t, x_{t+1}^\star \rangle \right]$$

$$= \psi_{T+1}(u) + \sum_{t=1}^{T} \left( F_t'(x_t^\star) - F_t'(x_{t+1}^\star) \right) - \sum_{t=1}^{T} \left( \psi_{t+1}(x_{t+1}^\star) - \psi_t(x_{t+1}^\star) \right)$$

$$= \psi_T(u) + \sum_{t=1}^{T} \left( F_t'(x_t^\star) - F_t'(x_{t+1}^\star) \right) - \frac{1}{2} \sum_{t=1}^{T} \langle z_t, x_t^\star \rangle^2$$

$$\leq \psi_T(u) + \sum_{t=1}^{T} \left( F_t'(x_t^\star) - F_t'(x_t') \right) - \frac{1}{2} \sum_{t=1}^{T} \langle z_t, x_t^\star \rangle^2 \,, \tag{36}$$

where we let $x_t' \in \arg\min_{x \in \mathbb{R}^n} F_t'(x)$; in particular, the first inequality is due to the fact that $F_{T+1}^\star(x_{T+1}^\star) \leq F_{T+1}^\star(u)$ and that $\psi_1$ is non-negative, whereas the last equality follows by definition of $\psi_t$ and $x_1^\star = 0$.

Consider now any term $F_t'(x_t^\star) - F_t'(x_t')$ in the sum after the last inequality and let $A_t = \eta_t I + \sum_{\tau=1}^{t} z_\tau z_\tau^\top$. Applying Lemma A.2 for $z_1 = x_t'$ and $z_2 = x_t^\star$ with $A = A_t$, we derive that

$$\|x_t^\star - x_t'\|_{A_t} \leq \frac{|y_t|}{2} \|z_t\|_{A_t^{-1}} \,. \tag{37}$$

We can now use this fact to show that

$$
\begin{aligned}
F_t'(x_t^\star) - F_t'(x_t') &\leq \langle \nabla F_t'(x_t^\star), x_t^\star - x_t' \rangle && \text{(convexity of } F_t') \\
&= \langle \nabla F_t^\star(x_t^\star) - y_t z_t, x_t^\star - x_t' \rangle && \text{(definition of } F_t') \\
&\leq y_t \langle z_t, x_t' - x_t^\star \rangle && \text{(first-order optimality)} \\
&\leq |y_t| \|z_t\|_{A_t^{-1}} \|x_t^\star - x_t'\|_{A_t} && \text{(Cauchy-Schwarz inequality)} \\
&\leq \frac{|y_t|^2}{2} \|z_t\|_{A_t^{-1}}^2 && \text{(Equation (37))} \\
&\leq \frac{Y^2}{2} \|z_t\|_{A_t^{-1}}^2 \,, \tag{38}
\end{aligned}
$$

where the last step is a consequence of $|y_t| \leq Y$ by Assumption 5.1. Further notice that $\|z_t\|_{A_t^{-1}}^2 \leq \|z_t\|_{A_{t-1}^{-1}}^2$ since $A_{t-1} \preceq A_t$, as well as

$$\|z_t\|_{A_t^{-1}}^2 \leq z_t^\top \left( \eta_t I + z_t z_t^\top \right)^{-1} z_t = z_t^\top \left( \frac{1}{\eta_t} I - \frac{z_t z_t^\top}{\eta_t^2 + \eta_t \|z_t\|_2^2} \right) z_t = \frac{\|z_t\|_2^2}{\eta_t} - \frac{\|z_t\|_2^4}{\eta_t^2 + \eta_t \|z_t\|_2^2} = \frac{\|z_t\|_2^2}{\eta_t + \|z_t\|_2^2} \leq 1 \,,$$

using the Sherman-Morrison formula at the first equality. Therefore, we show that the sum of the terms involving $F_t'$ is

$$\sum_{t=1}^{T} \left( F_t'(x_t^\star) - F_t'(x_t') \right) \leq \frac{Y^2}{2} \sum_{t=1}^{T} \|z_t\|_{A_t^{-1}}^2 \qquad \text{(Equation (38))}$$

$$\leq \frac{Y^2}{2} \sum_{t=1}^{T} \min \left\{ 1, \|z_t\|_{A_{t-1}^{-1}}^2 \right\}$$

$$\leq nY^2 \ln\left(1 + \frac{Z^2 T}{\eta_0 n}\right), \tag{39}$$

using Lemma 19.4 in Lattimore & Szepesvári (2020) at the last step. Then, combining together all these observations, we can bound $\mathrm{Reg}_T^\star(u)$ from above and obtain that

$$
\begin{aligned}
\mathrm{Reg}_T^\star(u) &\leq \sum_{t=1}^T \big(F_t'(x_t^\star) - F_t'(x_t')\big) + \psi_T(u) - \frac{1}{2}\sum_{t=1}^T \langle z_t, u\rangle^2 && \text{(Equations (35) and (36))} \\
&\leq nY^2 \ln\left(1 + \frac{Z^2 T}{\eta_0 n}\right) + \psi_T(u) - \frac{1}{2}\sum_{t=1}^T \langle z_t, u\rangle^2 && \text{(Equation (39))} \\
&= \frac{\eta_T}{2}\|u\|_2^2 + nY^2 \ln\left(1 + \frac{Z^2 T}{\eta_0 n}\right). && \text{(definition of } \psi_T\text{)} \tag{40}
\end{aligned}
$$

Let us now consider the drift term $\mathtt{Drift}_T$ from the decomposition in Equation (34). Define $\mathcal{T} = \{t \in [T] : f_t(\widetilde{x}_t) > f_t(x_t^\star)\}$ to be the rounds when $\widetilde{x}_t$ is worse than $x_t^\star$ with respect to the square loss $f_t$. Moreover, recall the definition of $\rho_t = \max_{\tau \in o_t}|y_\tau|$ as the threshold used for clipping in the definition of $\widetilde{x}_t$. By the convexity of $f_t$, we immediately have that

$$\mathtt{Drift}_T \leq \sum_{t\in\mathcal{T}}\big(f_t(\widetilde{x}_t) - f_t(x_t^\star)\big) \leq \sum_{t\in\mathcal{T}}\langle \nabla f_t(\widetilde{x}_t), \widetilde{x}_t - x_t^\star\rangle = \sum_{t\in\mathcal{T}}\big(\langle z_t, \widetilde{x}_t\rangle - y_t\big)\big(\langle z_t, \widetilde{x}_t\rangle - \langle z_t, x_t^\star\rangle\big). \tag{41}$$

Now, we distinguish the two following cases for any $t \in \mathcal{T}$:

- $f_t(\widetilde{x}_t) \leq f_t(x_t)$: thus, if $\langle z_t, \widetilde{x}_t\rangle \leq y_t$ it must be the case that $\langle z_t, x_t\rangle \leq \langle z_t, \widetilde{x}_t\rangle$, otherwise if $\langle z_t, \widetilde{x}_t\rangle > y_t$ then $\langle z_t, x_t\rangle \geq \langle z_t, \widetilde{x}_t\rangle$; in either case we have that

$$
\begin{aligned}
\big(\langle z_t, \widetilde{x}_t\rangle - y_t\big)\big(\langle z_t, \widetilde{x}_t\rangle - \langle z_t, x_t^\star\rangle\big) &\leq \big(\langle z_t, \widetilde{x}_t\rangle - y_t\big)\big(\langle z_t, x_t\rangle - \langle z_t, x_t^\star\rangle\big) \\
&\leq \big(|\rho_t| + |y_t|\big)|\langle z_t, x_t - x_t^\star\rangle| && \text{(triangle inequality, definition of } \widetilde{x}_t\text{)} \\
&\leq 2Y|\langle z_t, x_t - x_t^\star\rangle| && \text{(Assumption 5.1)} \\
&\leq 2Y\|z_t\|_{A_t^{-1}}\|x_t - x_t^\star\|_{A_t}. && \text{(Cauchy-Schwarz)} \tag{42}
\end{aligned}
$$

- $f_t(\widetilde{x}_t) > f_t(x_t)$: here it must be the case that $\widetilde{x}_t \neq x_t$, $y_t\langle z_t, \widetilde{x}_t\rangle \geq 0$, and $|y_t| > \rho_t$ (otherwise, clipping would have only decreased the square loss $f_t$); since $t \in \mathcal{T}$ implies that $|\langle z_t, x_t^\star\rangle - y_t| \leq |\langle z_t, \widetilde{x}_t\rangle - y_t|$, it follows that

$$
\begin{aligned}
\big(\langle z_t, \widetilde{x}_t\rangle - y_t\big)\big(\langle z_t, \widetilde{x}_t\rangle - \langle z_t, x_t^\star\rangle\big) &\leq |\langle z_t, \widetilde{x}_t\rangle - y_t|\big(|\langle z_t, \widetilde{x}_t\rangle - y_t| + |\langle z_t, x_t^\star\rangle - y_t|\big) && \text{(triangle inequality)} \\
&\leq 2\big(\langle z_t, \widetilde{x}_t\rangle - y_t\big)^2 \\
&= 2\big(|y_t| - |\langle z_t, \widetilde{x}_t\rangle|\big)^2 && (y_t\langle z_t, \widetilde{x}_t\rangle \geq 0) \\
&= 2\big(|y_t| - \rho_t\big)^2 && (|\langle z_t, \widetilde{x}_t\rangle| = \rho_t) \\
&< 2|y_t|^2. && (0 \leq \rho_t < |y_t|) \tag{43}
\end{aligned}
$$

Given the above remarks, let $\mathcal{T}_1 = \{t \in \mathcal{T} : f_t(\widetilde{x}_t) \leq f_t(x_t)\}$ be the subset of rounds in $\mathcal{T}$ when clipping does not worsen the value of $f_t$, and let $\mathcal{T}_2 = \mathcal{T} \setminus \mathcal{T}_1$ be the remaining rounds in $\mathcal{T}$. Then,

$$\mathtt{Drift}_T \leq \sum_{t\in\mathcal{T}}\big(\langle z_t, \widetilde{x}_t\rangle - y_t\big)\big(\langle z_t, \widetilde{x}_t\rangle - \langle z_t, x_t^\star\rangle\big) \leq 2Y\sum_{t\in\mathcal{T}_1}\|z_t\|_{A_t^{-1}}\|x_t - x_t^\star\|_{A_t} + 2\sum_{t\in\mathcal{T}_2}|y_t|^2. \tag{44}$$

At this point, for any round $t \in \mathcal{T}_1$ we are interested in understanding the behavior of $\|z_t\|_{A_t^{-1}}\|x_t - x_t^\star\|_{A_t}$. Applying Lemma A.2, we have that

$$\|x_t - x_t^\star\|_{A_t}^2 \leq \frac{1}{2}\sum_{\tau\in m_t} y_\tau\langle z_\tau, x_t^\star - x_t\rangle \leq \frac{1}{2}\sum_{\tau\in m_t} |y_\tau|\|z_\tau\|_{A_t^{-1}}\|x_t^\star - x_t\|_{A_t} \leq \frac{Y}{2}\sum_{\tau\in m_t}\|z_\tau\|_{A_t^{-1}}\|x_t^\star - x_t\|_{A_t},$$

where the second inequality follows by Cauchy-Schwarz, while the last one comes from Assumption 5.1. By rearranging terms in the previous inequality, we obtain that

$$\|x_t - x_t^\star\|_{A_t} \le \frac{Y}{2} \sum_{\tau \in m_t} \|z_\tau\|_{A_t^{-1}} . \tag{45}$$

Recall that we define $d_{\max}^{\le t} = \max_{\tau \le t} \min\{d_\tau, t - \tau\}$ as the maximum delay that has been perceived up to round $t$. Hence, we can now bound the sum relative to rounds in $\mathcal{T}_1$ from above as

$$2Y \sum_{t \in \mathcal{T}_1} \|z_t\|_{A_t^{-1}} \|x_t - x_t^\star\|_{A_t} \le Y^2 \sum_{t \in \mathcal{T}_1} \|z_t\|_{A_t^{-1}} \sum_{\tau \in m_t} \|z_\tau\|_{A_t^{-1}} \qquad \text{(Equation (45))}$$

$$\le Y^2 \sum_{t=1}^{T} \|z_t\|_{A_t^{-1}} \sum_{\tau \in m_t} \|z_\tau\|_{A_t^{-1}} .$$

If we now adopt Lemma C.1, we have that

$$\sum_{t=1}^{T} \|z_t\|_{A_t^{-1}} \sum_{\tau \in m_t} \|z_\tau\|_{A_t^{-1}} \le 2n d_{\max}^{\le T} \ln\left(1 + \frac{Z^2 T}{\eta_0 n}\right) ,$$

while at the same time we have

$$\sum_{t=1}^{T} \|z_t\|_{A_t^{-1}} \sum_{\tau \in m_t} \|z_\tau\|_{A_t^{-1}} \le Z^2 \sum_{t=1}^{T} \frac{|m_t|}{\eta_t} ,$$

where we used the fact that $\|z_s\|_{A_t^{-1}} \le \frac{Z^2}{\eta_t}$ for any $s \in [T]$. Thus, we have that

$$2Y \sum_{t \in \mathcal{T}_1} \|z_t\|_{A_t^{-1}} \|x_t - x_t^\star\|_{A_t} \le Y^2 \min\left\{2n d_{\max}^{\le T} \ln\left(1 + \frac{Z^2 T}{\eta_0 n}\right), Z^2 \sum_{t=1}^{T} \frac{|m_t|}{\eta_t}\right\} . \tag{46}$$

If we instead consider the sum over rounds in $\mathcal{T}_2$, it is possible to further bound it from above and relate it to the rounds for which the corresponding label does not belong to our estimate for the label range given by $\rho_t$. Indeed, if we let $\mathcal{R} = \{t \in [T] : |y_t| > \rho_t\}$ and given our previous remarks about $\mathcal{T}_2$, we have that $\mathcal{T}_2 \subseteq \mathcal{R}$. Now let $q_1 = \min\{\lceil \log_2 \rho_t \rceil : \rho_t > 0, t \in [T+1]\}$ and $q_2 = \lceil \log_2 \rho_{T+1} \rceil$. For convenience, define $\mathcal{I}_j = [2^j, 2^{j+1})$ for any $j \in \{q_1, \ldots, q_2\}$. Then, for any $t \in \mathcal{R}$, there exists $j_t \in \{q_1, \ldots, q_2\}$ such that $|y_t| \in \mathcal{I}_{j_t}$. Moreover, if we denote by $\nu_j \in [T+1]$ as the first time when $\rho_{\nu_j} \in \mathcal{I}_j$ for any $j \in \{q_1, \ldots, q_2\}$, we can further show that any $t \in \mathcal{R}$ has to be such that $t \in m_{\nu_{j_t}-1}$; if it were not the case, $y_t$ would have been observed before time $\nu_{j_t}$ which is a contradiction because $|y_t| > \rho_\tau$ for any $\tau < \nu_{j_t}$. All things considered, we can derive that

$$2 \sum_{t \in \mathcal{T}_2} |y_t|^2 \le 2 \sum_{t \in \mathcal{R}} |y_t|^2 \le 2 \sum_{j=q_1}^{q_2} \sum_{t \in m_{\nu_j-1}} |y_t|^2 \le 2 \sum_{j=q_1}^{q_2} 2^{2j} |m_{\nu_j-1}|$$

$$\le \sigma_{\max} \sum_{j=q_1}^{q_2} 2^{2j+1} \le \frac{8}{3} \sigma_{\max} 4^{q_2} \le \frac{32}{3} \sigma_{\max} \rho_{T+1}^2 \le 11 Y^2 \sigma_{\max} . \tag{47}$$

Combining all the results gathered so far, we can finally derive the overall regret bound as follows:

$$\text{Reg}_T(u) \le \text{Reg}_T^\star(u) + \text{Drift}_T$$

$$\le \frac{\eta_T}{2} \|u\|_2^2 + nY^2 \ln\left(1 + \frac{Z^2 T}{\eta_0 n}\right) + \text{Drift}_T \qquad \text{(Equation (40))}$$

$$\le \frac{\eta_T}{2} \|u\|_2^2 + nY^2 \ln\left(1 + \frac{Z^2 T}{\eta_0 n}\right) + 11 Y^2 \sigma_{\max} + 2Y \sum_{t \in \mathcal{T}_1} \|z_t\|_{A_t^{-1}} \|x_t - x_t^\star\|_{A_t} \qquad \text{(Equations (44) and (47))}$$

$$\tag{48}$$

$$\le \frac{\eta_T}{2}\|u\|_2^2 + nY^2\ln\left(1 + \frac{Z^2T}{\eta_0 n}\right) + 11Y^2\sigma_{\max}$$

$$+ Y^2\min\left\{2nd_{\max}\ln\left(1 + \frac{Z^2T}{\eta_0 n}\right), Z^2\sum_{t=1}^{T}\frac{|m_t|}{\eta_t}\right\}. \qquad \text{(Equation (46))} \qquad \square$$

The following corollary is a restatement of Corollary 5.3, which shows that we can further achieve a $\mathcal{O}(\min\{d_{\max}\ln T, \sqrt{d_{\text{tot}}}\})$ regret guarantee via an adaptive tuning of the learning rate of Algorithm 3 similar to the one adopted for Algorithm 2.

**Corollary 5.3.** *In the OLR problem with delayed labels under Assumption 5.1, Algorithm 3 with the adaptive learning rate $\eta_t = \gamma(\min\{a_t, b_t\} + 1)$, where $a_t$ and $b_t$ are defined in Equation* (13) *for any $\gamma > 0$ guarantees that*

$$\text{Reg}_T \le \frac{\gamma\|u\|_2^2}{2} + nY^2\ln\left(1 + \frac{Z^2T}{\gamma n}\right) + \mathcal{O}(\min\{Q_1, Q_2\}),$$

*where $Q_1 = (\gamma\|u\|_2^2 + Y^2)nd_{\max}\ln\left(1 + \frac{Z^2T}{\gamma n}\right)$ and $Q_2 = (\gamma Z\|u\|_2^2 + (Z+1)Y^2)\sqrt{d_{\text{tot}}}$.*

*Proof.* By performing a similar analysis as in the proof of Theorem 5.2 up to Equation (46), for any time threshold $\tau^\star \in [T]$ we can actually separately analyze the time ranges $\{1, \dots, \tau^\star\}$ and $\{\tau^\star + 1, \dots, T\}$ in an analogous way as in the proof of Corollary 4.2, and have a bound of the following form:

$$2Y\sum_{t\in\mathcal{T}_1}\|z_t\|_{A_t^{-1}}\|x_t - x_t^\star\|_{A_t} \le Y^2\left(2nd_{\max}^{\le\tau^\star}\ln\left(1 + \frac{Z^2T}{\eta_0 n}\right) + Z^2\sum_{t=\tau^\star+1}^{T}\frac{|m_t|}{\eta_t}\right). \tag{49}$$

Then, we use an adaptive tuning of the learning rate in a similar way as performed for the proof of Corollary 4.2. In particular, we define

$$a_t = 2nd_{\max}^{\le t}\ln\left(1 + \frac{Z^2T}{\gamma n}\right) \quad \text{and} \quad b_t = Z\sqrt{\sum_{s=1}^{t}|m_s|},$$

and, for any $\gamma > 0$, we set $\eta_0 = \gamma$ and $\eta_t = \gamma(\min\{a_t, b_t\} + 1)$ for any $t \ge 1$. First, when $a_T \le b_T$ we have that

$$\text{Reg}_T(u) \le \frac{\eta_T}{2}\|u\|_2^2 + nY^2\ln\left(1 + \frac{Z^2T}{\gamma n}\right) + Y^2\left(11\sigma_{\max} + 2nd_{\max}\ln\left(1 + \frac{Z^2T}{\gamma n}\right)\right) \quad \text{(Equations (46) and (48))}$$

$$\le \frac{\|u\|_2^2}{2}\eta_T + nY^2\ln\left(1 + \frac{Z^2T}{\gamma n}\right) + Y^2 d_{\max}\left(11 + 2n\ln\left(1 + \frac{Z^2T}{\gamma n}\right)\right) \quad (\sigma_{\max} \le d_{\max})$$

$$\le \frac{\gamma\|u\|_2^2}{2} + nY^2\ln\left(1 + \frac{Z^2T}{\gamma n}\right) + 11Y^2 d_{\max} + (\gamma\|u\|_2^2 + 2Y^2)nd_{\max}\ln\left(1 + \frac{Z^2T}{\gamma n}\right)$$

$$\le \frac{\gamma\|u\|_2^2}{2} + nY^2\ln\left(1 + \frac{Z^2T}{\gamma n}\right) + (\gamma\|u\|_2^2 + 13Y^2)nd_{\max}\ln\left(1 + \frac{Z^2T}{\gamma n}\right).$$

On the contrary, when $a_T > b_T$, we let $\tau^\star$ be the last round such that $a_{\tau^\star} \le b_{\tau^\star}$ and show that

$$\text{Reg}_T(u) \le \frac{\|u\|_2^2}{2}\eta_T + nY^2\ln\left(1 + \frac{Z^2T}{\gamma n}\right) + 11Y^2\sigma_{\max} + Y^2\left(2nd_{\max}^{\le\tau^\star}\ln\left(1 + \frac{Z^2T}{\gamma n}\right) + Z^2\sum_{t=\tau^\star+1}^{T}\frac{|m_t|}{\eta_t}\right)$$

$$\text{(Equations (48) and (49))}$$

$$\le \frac{\|u\|_2^2}{2}\eta_T + nY^2\ln\left(1 + \frac{Z^2T}{\gamma n}\right) + 11Y^2\sigma_{\max} + ZY^2\left(\sqrt{\sum_{t=1}^{\tau^\star}|m_t|} + Z\sum_{t=\tau^\star+1}^{T}\frac{|m_t|}{\eta_t}\right) \quad (a_{\tau^\star} \le b_{\tau^\star})$$

$$\leq \frac{\|u\|_2^2}{2}\eta_T + nY^2\ln\left(1 + \frac{Z^2T}{\gamma n}\right) + 11Y^2\sigma_{\max} + \frac{ZY^2}{\gamma}\left(\sqrt{\sum_{t=1}^{\tau^\star}|m_t|} + \sum_{t=\tau^\star+1}^{T}\frac{|m_t|}{\sqrt{\sum_{s=1}^{t}|m_s|}}\right)$$

(definition of $\eta_t$)

$$\leq \frac{\|u\|_2^2}{2}\eta_T + nY^2\ln\left(1 + \frac{Z^2T}{\gamma n}\right) + 11Y^2\sigma_{\max} + ZY^2\left(\sqrt{\sum_{t=1}^{\tau^\star}|m_t|} + 2\sqrt{\sum_{t=\tau^\star+1}^{T}|m_s|}\right)$$

(Orabona (2025, Lemma 4.13))

$$\leq \frac{\|u\|_2^2}{2}\eta_T + nY^2\ln\left(1 + \frac{Z^2T}{\gamma n}\right) + 11Y^2\sigma_{\max} + 2ZY^2\sqrt{2d_{\text{tot}}}$$

$$\leq \frac{\|u\|_2^2}{2}\eta_T + nY^2\ln\left(1 + \frac{Z^2T}{\gamma n}\right) + 2(11+Z)Y^2\sqrt{2d_{\text{tot}}}$$

(Lemma A.7)

$$\leq \frac{\gamma\|u\|_2^2}{2}\left(1 + Z\sqrt{d_{\text{tot}}}\right) + nY^2\ln\left(1 + \frac{Z^2T}{\gamma n}\right) + 2(11+Z)Y^2\sqrt{2d_{\text{tot}}}\ .$$

(definition of $\eta_T$)

Considering the conditions in each of the two cases together with the definitions of $a_t$ and $b_t$, this concludes the proof. $\qquad\square$

## E. Online mirror descent for delayed OCO with strongly convex losses

In this section, we prove that the following online mirror descent (OMD) algorithm achieves a regret guarantee whose dependence on the delays is of order $\min\{\sigma_{\max}\ln T, \sqrt{d_{\text{tot}}}\}$, similarly to Algorithm 1. To be precise, an OMD-based algorithm which handles delays was initially proposed by Wu et al. (2024) in their Algorithm 6. However, Wu et al. (2024) only manage to show that this algorithm achieves regret $\mathcal{O}\left(\frac{d_{\max}(G^2+D)}{\lambda}\ln T + \frac{d_{\max}G}{\lambda^2}\right)$ under Assumptions 2.3 and 2.4. Here, we report its pseudocode in Algorithm 4 and we provide an improved regret analysis for it. Not only do we provide a significantly better guarantee, but we also manage to lift Assumption 2.4 and only require the boundedness of the gradient norms via Assumption 2.3. The key to achieve these improvements simultaneously is a fundamentally different and more careful regret analysis.

---

**Algorithm 4** Delayed OMD for strongly convex functions

---

**input** strong convexity parameter $\lambda > 0$, learning rates $\eta_t = \frac{2}{t\lambda}$ for all $t \in [T]$
**initialize** $x_1 \in \mathcal{X}$
1: **for** $t = 1, 2, \dots$ **do**
2:     Play $x_t$
3:     Receive $g_\tau = \nabla f_\tau(x_\tau)$ for all $\tau \in o_{t+1} \setminus o_t$
4:     Update $x_{t+1} = \arg\min_{x\in\mathcal{X}} \sum_{\tau \in o_{t+1}\setminus o_t} \langle g_\tau, x \rangle + \frac{1}{\eta_t}\|x - x_t\|_2^2$.
5: **end for**

---

**Theorem E.1.** *Assume that $f_1, \dots, f_T$ are $\lambda$-strongly convex functions with respect to the Euclidean norm $\|\cdot\|_2$. Then, under Assumption 2.3, Algorithm 4 guarantees*

$$\text{Reg}_T = \mathcal{O}\left(\frac{G^2}{\lambda}\left(\ln T + \min\left\{\sigma_{\max}\ln T, \sqrt{d_{\text{tot}}}\right\}\right)\right)\ .$$

*Proof.* We begin with a decomposition of the regret that, similarly to the proof of Theorem 3.1, leverages the strong convexity of losses $f_1, \dots, f_T$ and attempts to isolate the discrepancy in the information available to the learner because of the delayed gradients. However, this decomposition differs from the one in Theorem 3.1 since the algorithm updates its predictions differently via mirror descent. Our approach follows the idea of framing such an information discrepancy via optimism (Flaspohler et al., 2021). For notational convenience, define $\widetilde{g}_1 = 0$ and $\widetilde{g}_{t+1} = \widetilde{g}_t + \sum_{\tau \in o_{t+1}\setminus o_t} g_\tau - g_t$ for any $t \geq 1$. Note that, by definition, each $\widetilde{g}_t$ is equal to

$$\widetilde{g}_t = \sum_{\tau=1}^{t-1}\left(\widetilde{g}_{\tau+1} - \widetilde{g}_\tau\right) = \sum_{s=1}^{t-1}\left(\sum_{\tau\in o_{s+1}\setminus o_s} g_\tau - g_s\right) = \sum_{s\in o_t} g_s - \sum_{s=1}^{t-1} g_s = -\sum_{s\in m_t} g_s \tag{50}$$

and consequently $\widetilde{g}_{T+1} = 0$ since $m_{T+1} = \emptyset$. This definition of $\widetilde{g}_t$ allows to rewrite the "linearized" regret as

$$\sum_{t=1}^{T}\langle g_t, x_t - u\rangle = \sum_{t=1}^{T}\left\langle \sum_{\tau \in o_{t+1}\setminus o_t} g_\tau, x_t - u\right\rangle + \sum_{t=1}^{T}\langle \widetilde{g}_t - \widetilde{g}_{t+1}, x_t\rangle \qquad (51)$$

and to have that, for every round $t$,

$$\left\langle \sum_{\tau \in o_{t+1}\setminus o_t} g_\tau, x_t - x_{t+1}\right\rangle = \langle g_t - \widetilde{g}_t + \widetilde{g}_{t+1}, x_t - x_{t+1}\rangle = \langle g_t - \widetilde{g}_t, x_t - x_{t+1}\rangle + \langle \widetilde{g}_{t+1}, x_t - x_{t+1}\rangle. \qquad (52)$$

Moreover, according to the standard regret analysis of OMD (Lemma A.4), we know that

$$\left\langle \sum_{\tau \in o_{t+1}\setminus o_t} g_\tau, x_t - u\right\rangle \leq \frac{1}{\eta_t}\left(\|u - x_t\|_2^2 - \|u - x_{t+1}\|_2^2 - \|x_t - x_{t+1}\|_2^2\right) + \left\langle \sum_{\tau \in o_{t+1}\setminus o_t} g_\tau, x_t - x_{t+1}\right\rangle. \qquad (53)$$

The above observations then make it possible to bound the first sum in the right-hand side of Equation (51) as

$$\sum_{t=1}^{T}\left\langle \sum_{\tau \in o_{t+1}\setminus o_t} g_\tau, x_t - u\right\rangle \leq \sum_{t=1}^{T}\frac{1}{\eta_t}\left(\|u - x_t\|_2^2 - \|u - x_{t+1}\|_2^2 - \|x_t - x_{t+1}\|_2^2\right)$$

$$+ \sum_{t=1}^{T}\left\langle \sum_{\tau \in o_{t+1}\setminus o_t} g_\tau, x_t - x_{t+1}\right\rangle \qquad \text{(Equation (53))}$$

$$= \sum_{t=1}^{T}\frac{1}{\eta_t}\left(\|u - x_t\|_2^2 - \|u - x_{t+1}\|_2^2 - \|x_t - x_{t+1}\|_2^2\right)$$

$$+ \sum_{t=1}^{T}\langle g_t - \widetilde{g}_t, x_t - x_{t+1}\rangle + \sum_{t=1}^{T}\langle \widetilde{g}_{t+1}, x_t - x_{t+1}\rangle \qquad \text{(Equation (52))}$$

$$= \sum_{t=1}^{T}\frac{1}{\eta_t}\left(\|u - x_t\|_2^2 - \|u - x_{t+1}\|_2^2 - \|x_t - x_{t+1}\|_2^2\right)$$

$$+ \sum_{t=1}^{T}\langle g_t - \widetilde{g}_t, x_t - x_{t+1}\rangle + \sum_{t=1}^{T}\langle \widetilde{g}_{t+1} - \widetilde{g}_t, x_t\rangle$$

$$+ \langle \widetilde{g}_1, x_1\rangle - \langle \widetilde{g}_{T+1}, x_{T+1}\rangle$$

$$= \sum_{t=1}^{T}\frac{1}{\eta_t}\left(\|u - x_t\|_2^2 - \|u - x_{t+1}\|_2^2 - \|x_t - x_{t+1}\|_2^2\right)$$

$$+ \sum_{t=1}^{T}\langle g_t - \widetilde{g}_t, x_t - x_{t+1}\rangle + \sum_{t=1}^{T}\langle \widetilde{g}_{t+1} - \widetilde{g}_t, x_t\rangle, \qquad (54)$$

where the second equality follows by carefully rearranging the terms in the sum $\sum_{t=1}^{T}\langle \widetilde{g}_{t+1}, x_t - x_{t+1}\rangle$, while the last equality is due to $\widetilde{g}_1 = \widetilde{g}_{T+1} = 0$ by definition.

At this point, we can rewrite the regret in the following way:

$$\text{Reg}_T(u) = \sum_{t=1}^{T}\left(f_t(x_t) - f_t(u)\right)$$

$$\leq \sum_{t=1}^{T}\langle g_t, x_t - u\rangle - \frac{\lambda}{2}\sum_{t=1}^{T}\|x_t - u\|_2^2$$

$$= \sum_{t=1}^{T}\left\langle \sum_{\tau \in o_{t+1}\setminus o_t} g_\tau, x_t - u\right\rangle + \sum_{t=1}^{T}\langle \widetilde{g}_t - \widetilde{g}_{t+1}, x_t\rangle - \frac{\lambda}{2}\sum_{t=1}^{T}\|x_t - u\|_2^2 \qquad \text{(Equation (51))}$$

$$\leq \sum_{t=1}^{T} \frac{\|u - x_t\|_2^2 - \|u - x_{t+1}\|_2^2 - \|x_t - x_{t+1}\|_2^2}{\eta_t} + \sum_{t=1}^{T} \langle g_t - \widetilde{g}_t, x_t - x_{t+1} \rangle - \frac{\lambda}{2} \sum_{t=1}^{T} \|x_t - u\|_2^2$$

(Equation (54))

$$= \sum_{t=1}^{T} \left( \frac{\|u - x_t\|_2^2 - \|u - x_{t+1}\|_2^2}{\eta_t} - \frac{\lambda}{2} \sum_{t=1}^{T} \|x_t - u\|_2^2 \right) + \sum_{t=1}^{T} \left( \langle g_t - \widetilde{g}_t, x_t - x_{t+1} \rangle - \frac{\|x_t - x_{t+1}\|_2^2}{\eta_t} \right)$$

$$= \frac{\lambda}{2} \sum_{t=1}^{T} \left( (\|x_t - u\|_2^2 - \|x_{t+1} - u\|_2^2)t - \|x_t - u\|_2^2 \right) + \sum_{t=1}^{T} \left( \langle g_t - \widetilde{g}_t, x_t - x_{t+1} \rangle - \frac{\|x_t - x_{t+1}\|_2^2}{\eta_t} \right)$$

(definition of $\eta_t$)

$$= -\frac{\lambda T}{2} \|x_{T+1} - u\|_2^2 + \sum_{t=1}^{T} \left( \langle g_t - \widetilde{g}_t, x_t - x_{t+1} \rangle - \frac{\|x_t - x_{t+1}\|_2^2}{\eta_t} \right)$$

$$\leq \sum_{t=1}^{T} \left( \langle g_t - \widetilde{g}_t, x_t - x_{t+1} \rangle - \frac{\|x_t - x_{t+1}\|_2^2}{\eta_t} \right) , \tag{55}$$

where the first inequality holds because of the $\lambda$-strong convexity of $f_t$.

We now focus on the right-hand side of Equation (55). Applying Lemma A.2, we can bound from above the distance between subsequent iterates:

$$\|x_t - x_{t+1}\|_2 \leq \eta_t \|g_t + \widetilde{g}_{t+1} - \widetilde{g}_t\|_2 = \eta_t \left\| \sum_{\tau \in o_{t+1} \setminus o_t} g_\tau \right\|_2 \leq G\eta_t \left( |o_{t+1}| - |o_t| \right) , \tag{56}$$

where the last inequality follows by jointly using the triangle inequality, the bound on the gradient norm (Assumption 2.3), and the fact that $o_t \subseteq o_{t+1}$.

What remains to analyze now is the distance $\|g_t - \widetilde{g}_t\|_2$, and a direct calculation allows us to show that

$$\|g_t - \widetilde{g}_t\|_2 = \left\| g_t + \sum_{\tau \in m_t} g_\tau \right\|_2 \leq G(|m_t| + 1) , \tag{57}$$

again by using the triangle inequality and Assumption 2.3.

Applying Lemma A.5 with Equation (56), we show that the each term of the sum in the right-hand side of Equation (55) satisfies

$$\langle g_t - \widetilde{g}_t, x_t - x_{t+1} \rangle - \frac{\|x_t - x_{t+1}\|_2^2}{\eta_t} \leq \min \left\{ G\eta_t \|g_t - \widetilde{g}_t\|_2 (|o_{t+1}| - |o_t|), \eta_t \|g_t - \widetilde{g}_t\|_2^2 \right\} . \tag{58}$$

Therefore, starting from Equation (55), we are able to derive the final regret bound:

$$\text{Reg}_T(u) \leq \sum_{t=1}^{T} \eta_t \|g_t - \widetilde{g}_t\|_2 \|g_t + \widetilde{g}_{t+1} - \widetilde{g}_t\|_2 \qquad \text{(Equations (55) and (58))}$$

$$= \frac{2G}{\lambda} \sum_{t=1}^{T} \frac{\|g_t - \widetilde{g}_t\|_2 (|o_{t+1}| - |o_t|)}{t} \qquad \text{(definition of } \eta_t)$$

$$\leq \frac{2G^2}{\lambda} \sum_{t=1}^{T} \frac{(|m_t| + 1)(|o_{t+1}| - |o_t|)}{t} . \qquad \text{(Equation (57))}$$

Crucially, what remains to analyze is the sum in the right-hand side of the above inequality. We can first show that

$$\sum_{t=1}^{T} \frac{(|m_t| + 1)(|o_{t+1}| - |o_t|)}{t} \leq (\sigma_{\max} + 1) \sum_{t=1}^{T} \frac{|o_{t+1}| - |o_t|}{t} \qquad \text{(definition of } \sigma_{\max})$$

$$\leq (\sigma_{\max} + 1) \sum_{t=1}^{T} \frac{|o_{t+1}| - |o_t|}{|o_{t+1}|} \qquad (o_{t+1} \subseteq [t])$$

$$= (\sigma_{\max} + 1) \sum_{t=1}^{T} \frac{(|o_{t+1}| - |o_t|)}{\sum_{s=1}^{t}(|o_{s+1}| - |o_s|)}$$

$$\leq (\sigma_{\max} + 1)(1 + \ln T) , \tag{59}$$

where the last inequality follows by Orabona (2025, Lemma 4.13) and the fact that $\sum_{t=1}^{T}(|o_{t+1}| - |o_t|) = |o_{T+1}| = T$. Second, we can also bound such a sum in an alternative way:

$$\sum_{t=1}^{T} \frac{(|m_t| + 1)(|o_{t+1}| - |o_t|)}{t} = \sum_{t=1}^{T} \frac{|m_t|(|o_{t+1}| - |o_t|)}{t} + \sum_{t=1}^{T} \frac{(|o_{t+1}| - |o_t|)}{t}$$

$$\leq \sum_{t=1}^{T} \frac{|m_t|(|o_{t+1}| - |o_t|)}{t} + \sum_{t=1}^{T} \frac{(|o_{t+1}| - |o_t|)}{\sum_{s=1}^{t}(|o_{s+1}| - |o_s|)} \qquad \text{(definition of } o_t\text{)}$$

$$\leq \sum_{t=1}^{T} \frac{|m_t|(|o_{t+1}| - |o_t|)}{t} + \ln T + 1$$

$$\leq \sum_{t=1}^{T} \frac{|m_t|(t - |m_{t+1}| - (t - 1 - |m_t|))}{t} + \ln T + 1 \qquad (|o_t| + |m_t| = t - 1 \text{ for all } t)$$

$$= \sum_{t=1}^{T} \frac{|m_t|(1 + |m_t| - |m_{t+1}|)}{t} + \ln T + 1$$

$$= \sum_{t=1}^{T} \frac{|m_t|}{t} + |m_1|^2 - \frac{|m_T||m_{T+1}|}{t} + \sum_{t=2}^{T} \left( \frac{|m_t|^2}{t} - \frac{|m_{t-1}||m_t|}{t - 1} \right) + \ln T + 1$$

$$= \sum_{t=1}^{T} \frac{|m_t|}{t} + \sum_{t=2}^{T} \left( \frac{|m_t|^2}{t} - \frac{|m_{t-1}||m_t|}{t - 1} \right) + \ln T + 1 \qquad \text{(definition of } m_t\text{)}$$

$$\leq \sum_{t=1}^{T} \frac{|m_t|}{t} + \sum_{t=2}^{T} \left( \frac{(|m_{t-1}| + 1)|m_t|}{t - 1} - \frac{|m_{t-1}||m_t|}{t - 1} \right) + \ln T + 1$$

$$(m_{t+1} \subseteq m_t \cup \{t\} \text{ for all } t)$$

$$\leq \sum_{t=1}^{T} \frac{|m_t|}{t} + \ln T + 1$$

$$\leq 2\sqrt{d_{\text{tot}}} + \ln T + 1 ,$$

where the last inequality follows by Equation (22). Combing the above two inequalities, we finally obtain

$$\text{Reg}_T(u) \leq \frac{2G^2}{\lambda}(1 + \ln T) + \frac{2G^2}{\lambda} \min \left\{ \sigma_{\max}(1 + \ln T), 2\sqrt{d_{\text{tot}}} \right\}$$

$$= \mathcal{O}\left( \frac{G^2}{\lambda} \left( \ln T + \min \left\{ \sigma_{\max} \ln T, \sqrt{d_{\text{tot}}} \right\} \right) \right) . \qquad \square$$

## F. Additional experiments

We consider a real-world dataset $mg\_scale$ from the LIBSVM repository (Chang & Lin, 2011). This dataset has 1385 samples and each sample has 6 features with values in $[-1, 1]$ and a label in $[0, 2]$. The experimental setup, including constructions of losses and delays, follows what already done for the experiments in Section 6. Figure 2 shows a similar behaviour of the algorithms as already shown in Section 6.

We also designed a non-stationary environment as follows. The generation processes for the feature vectors, as well as the definition of the loss function, remain the same as the environment in Section 6. However, we modified the generation of the

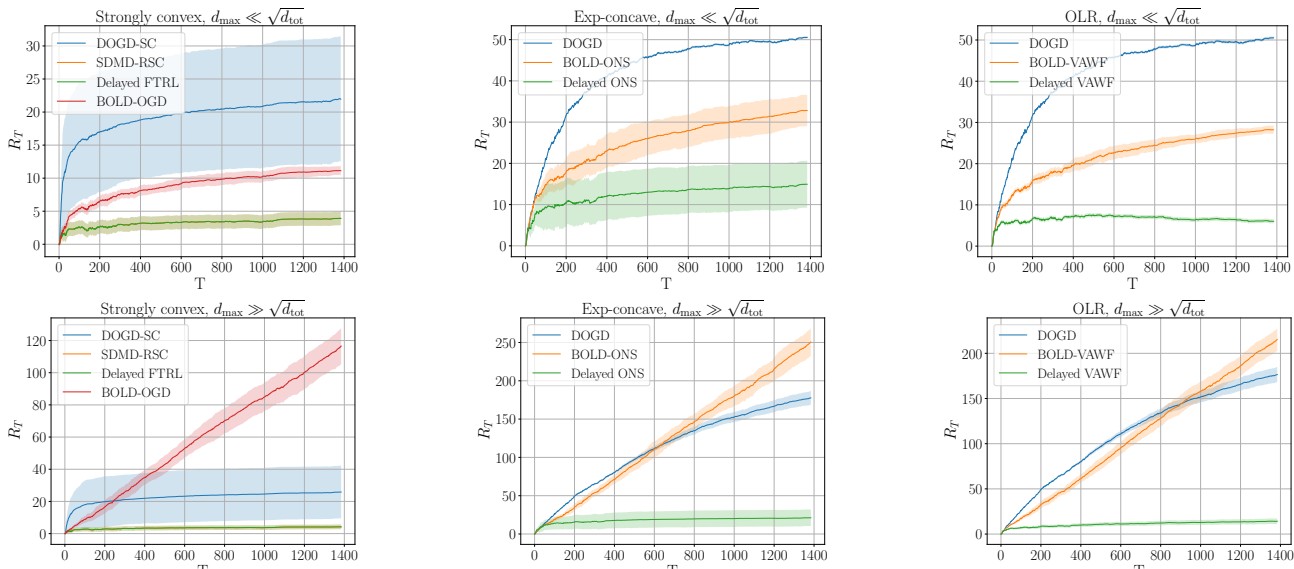

*Figure 2.* Comparison with relevant baselines. The shaded areas consider a range centered around the mean with half-width corresponding to the empirical standard deviation over 20 repetitions.

label $y_t$:

$$y_t = \langle z_t, \theta_t \rangle + \epsilon_t , \tag{60}$$

where the latent vector $\theta_t$ alternates every 30 rounds between the two vectors $\mathbf{1}$ and $\mathbf{0}$. This periodic change introduces non-stationarity, reflecting scenarios where the optimal action shifts over time. The delay $d_t$ is independently sampled from a distribution that alternates every 30 rounds between a geometric distribution with success probability $T^{-1/3}$ and a uniform distribution over the set $\{0, 1, \ldots, 5\}$. Additionally, we also modify the noise term $\epsilon_t$ inspired by Xu & Zeevi (2023). Specifically, we flatten an abstract art piece by Jackson Pollock and take consecutive grayscale values in $[0, 1]$ as the noise $\epsilon_t$. Figure 3 shows that our algorithms again perform the best among all the benchmark algorithms.

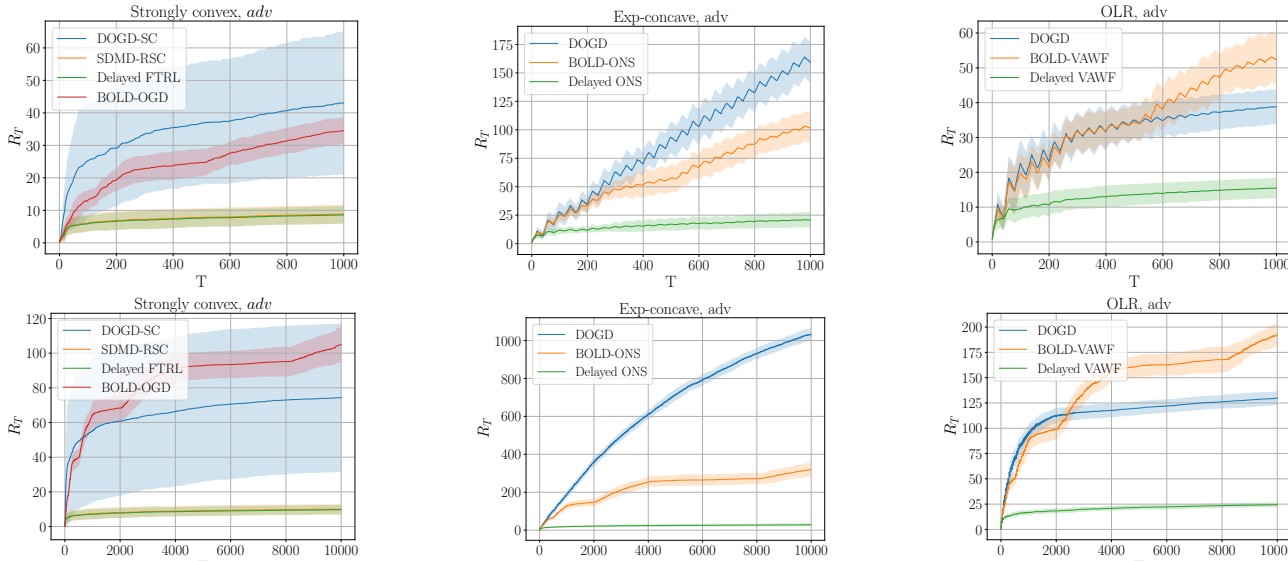

*Figure 3.* Comparison with relevant baselines. The shaded areas consider a range centered around the mean with half-width corresponding to the empirical standard deviation over 20 repetitions. The top plots correspond to $T = 1000$, while the bottom plots correspond to $T = 10000$.

