# OpenReview forum: "Exploiting Curvature in Online Convex Optimization with Delayed Feedback"
_ICML.cc/2025/Conference — ICML 2025 poster_

### Official Review · Reviewer_htSw · 2025-03-12

**Overall Recommendation:** 3

**Summary:**

This paper studies online learning with delayed feedback under curved loss functions. Specifically, for strongly convex functions, the proposed FTRL method achieves a regret bound of $O(\min\{\sigma\_{\max} \ln T, \sqrt{d}\_{\text{tot}}\})$, where $\sigma\_{\max}$ denotes the maximum number of missing data. This result improves upon the best-known bound of $O(d\_{\max} \ln T)$ for strongly convex functions. For exp-concave functions, the paper establishes that a regret bound of $O(\min\\{d\_{\max} n \ln T, \sqrt{d}\_{\text{tot}}\\})$ is achievable. Additionally, the similar idea extends to the VAW algorithm for online least-squares regression.

**Claims And Evidence:**

To me, the claim made in the paper is well supported.

**Essential References Not Discussed:**

It appears that the relevant works have been cited. However, I am unsure whether the method of Wu et al. (2024) can be extended to the exp-concave loss setting through a specific choice of the Bregman divergence. Additionally, Wu et al. (2024) also studied FTRL-based methods. I believe it would be beneficial to provide a more detailed comparison highlighting the differences between the proposed methods and the previous one.

**Experimental Designs Or Analyses:**

- It appears that the results of the BOLD-OGD methods for strongly convex functions are not presented in Figure (a) and Figure (d).
- In the strongly convex case, although the proposed methods have a clearly better regret bound than SDMD-RSC, their empirical performance remains similar in both cases: $d_{\max} \ll \sqrt{d_{\text{tot}}}$ and $d_{\max} \gg \sqrt{d_{\text{tot}}}$. It is unclear whether this is due to shortcomings in the experimental setup or if the analysis of the previous method is overly loose.
- For exp-concave loss functions, I believe it would be fair to compare with methods that achieve the $\sqrt{d_{\text{tot}}}$ bound to assess whether the curvature of the loss function really helps for learning with delayed feedback.
- It seems that the method by Wu et al. (2024) applies to relatively strongly convex functions. I am unsure whether their approach can also be extended to the exp-concave case by selecting the function in the Bregman divergence as $\frac{1}{2} \Vert x \Vert^2_{\nabla f_t \nabla f_t^\top}$ within their framework.

**Methods And Evaluation Criteria:**

Yes, regret is typically used as a performance measure for online learning methods.

**Other Comments Or Suggestions:**

Please see above.

**Other Strengths And Weaknesses:**

**Strengths:**
+ The paper presents a new algorithm for learning with delayed feedback, offering an improved bound.
+ The method and proof are clear and well-structured.
+ The empirical results validate the effectiveness of the proposed method.

**Weaknesses:**
- It is unclear how significant the improvement of the proposed method is. Theoretically, it would be beneficial for the authors to provide concrete examples illustrating when and why the improvement is substantial. Experimentally, the proposed method appears to perform similarly to the previous approach for strongly convex functions.

**Questions For Authors:**

- Please refer to the first three points in the "Experimental Designs or Analyses" part.
- Could you discuss whether the method of Wu et al. (2024) can also be applied to the exp-concave case by selecting an appropriate function in the Bregman divergence?
- Could you provide some concrete examples where the improvement of the proposed method is significant, both theoretically and empirically, for strongly convex functions?

**Relation To Broader Scientific Literature:**

The paper presents an algorithm for delayed feedback within the OCO framework with curved loss, with its main contribution being introducing methods with improved bounds.

**Theoretical Claims:**

I have gone through the proof for the strongly convex function and did not find any issues. The proofs for the exp-concave loss and the VAW method also appear correct.

---

> ### Author Rebuttal · Authors · 2025-04-01
>
> **Q:** Missing results of BOLD-OGD for strongly convex functions in Figures (a) and (b).
>
> **A:** To address the request of a more comprehensive empirical comparison, we will also include the performance of BOLD-OGD in the strongly convex setting. We originally omitted BOLD-OGD because it is a typically inefficient algorithm (needs to run $\sigma_{\max}+1$ independent instances of some base algorithm, here OGD), and for the strongly convex setting (contrarily to the exp-concave and OLR settings) we already have specific benchmarks from prior work (Wan et al., 2022; Wu et al., 2024) to compare against which are more practical and usually show an improved empirical performance compared to the black-box reduction via BOLD.
> You can find these new plots [here](https://anonymous.4open.science/r/Supplementary-experiments-EF2D), showing our algorithm outperforms BOLD-OGD too.
>
> **Q:** Similar performance to SDMD-RSC in the strongly convex case; examples showing the improvement by the proposed methods under strong convexity.
>
> **A:** In fact, we do prove in Appendix F that delayed OMD (originally from Wu et al., 2024) achieves the same optimal regret bound as our delayed FTRL under strong convexity, which is one of the contributions of our work. We remark that showing this result required a non-negligible effort as our analysis is **radically different** from the one by Wu et al. (2024), which was a crucial difference in deriving the improved guarantees. Specifically, as mentioned in our Section F and Section 2 as well as in our response to Reviewers jw1R and zZM9, our bound is essentially optimal, while the $O(\frac{G^2+D}{\lambda}(d_{\max}+1)\ln T + \frac{G}{\lambda^2}(d_{\max}+1))$ bound proved in Wu et al. (2024) has some disadvantages including: (1) a multiplicative dependence on the diameter $D$ of the action set $\mathcal{X}$; (2) no robustness to the $\sqrt{d_{tot}}$ regime; (3) a worse dependence on the strong convexity parameter $\lambda$; (4) a worse $d_{\max}$ dependence. In turn, our regret bound for delayed OMD improves upon all these points and recovers the optimal $O(\frac{G^2}{\lambda}\ln T)$ regret in the no-delay setting under strong convexity. Our improved analysis also explains why the empirical performances of the two algorithms we studied have a similar performance in our experiments for the strongly convex case.
>
> **Q:** Empirical comparison with methods achieving $\sqrt{d_{tot}}$ regret in the exp-concave case.
>
> **A:** We remark that DOGD is one such algorithm and we do compare our method to DOGD in our experiments. The results are shown in Figures (b) and (e), and that our algorithm indeed outperforms DOGD in both cases.
>
> **Q:** Possibly extending results of Wu et al. (2024) to the exp-concave setting.
>
> **A:** Based on our understanding, extending the results of Wu et al. (2024) to the exp-concave setting is highly nontrivial. Wu et al. (2024) focuses exclusively on OMD and FTRL algorithms with a fixed regularizer $\psi$, and their analysis critically relies on this assumption. In contrast, analyzing exp-concave functions typically requires handling time-varying or data-dependent regularizers (required as illustrated by the reviewer themselves, and addressed by our Algorithm 2), for which Wu et al.'s techniques do not directly apply. Addressing this challenge under delayed feedback is one of the key technical contributions of our work. We will better highlight this difference and our contribution with respect to this aspect in our next revision.

---

### Official Review · Reviewer_zZM9 · 2025-03-12

**Overall Recommendation:** 3

**Summary:**

This paper investigates various types of loss functions in the context of Online Convex Optimization (OCO) with delayed feedback and proposes a variant of Follow-The-Regularized-Leader (FTRL) to improve upon previous results. Firstly, it slightly enhances the existing regret bound for strongly convex loss functions. Secondly, it provides the first theoretical guarantee for exp-concave loss functions. Additionally, the authors analyze online linear regression with delayed feedback. Finally, experimental results confirm the effectiveness of their proposed approach.

**Claims And Evidence:**

Yes, it improves the previous results.

**Essential References Not Discussed:**

No

**Experimental Designs Or Analyses:**

The experiment lacks a description of the parameter settings for the baseline methods.

In the experiments, the learning rate for DONS appears to be set differently from what is stated in the theorem. Could you clarify the rationale behind this choice? I am concerned that the experimental results may be highly dependent on the specific learning rate setting, which could impact the generality and robustness of your results.

**Methods And Evaluation Criteria:**

Yes

**Other Comments Or Suggestions:**

- Typo Line 929  delete $d$.

**Other Strengths And Weaknesses:**

**Strengths**

- The paper is well-written and easy to follow.
- The authors provide the code for the experiments.
- The idea for exp-concave loss functions is interesting, although there are some unclear presentation (see Theoretical Claims).


**Weakness**

- I think the improvement for strongly convex loss functions is somewhat limited.
- The authors need to validate their findings with experiments on both real-world tasks.
- The results of OLR appear to be a combination of the results from ONS and Vovk-Azoury-Warmuth (VAW), which further limits its contribution.

**Questions For Authors:**

- I concern about the assumption $t+d_t \leq T$, as it does not appear in previous works [1]. Could you clarify the role this assumption plays in your analysis?


- The authors claim that their regret guarantee does not depend on the diameter of the domain. This is primarily attributed to the inclusion of an additional $Drift_T$ term. However, I think the constant improvement provided by this term does not constitute a significant contribution to the overall result.


[1] Online learning with adversarial delays. Advances in neural information processing systems, 2015.

**Relation To Broader Scientific Literature:**

This work is relative to [1] and [2], and claims answering an open question proposed in [1].

[1] Online strongly convex optimization with unknown delays. Machine Learning, 2022.

[2] Online sequential decisionmaking with unknown delays. In Proceedings of the ACM on Web Conference 2024, 2024.

**Theoretical Claims:**

There is an issue with the results presented in the abstract and introduction. The authors assume the delay $d_t \geq 0$.  Under the no-delay setting ($d_t = 0$), their regret bound in the Table~1 (i.e., $\min \\{\sigma_{\max}\ln T, \sqrt{d_{tot}} \\}$) reduces to  $O(1)$.
This is because of the discarded term $O(\ln T)$. The complete regret bound should be presented as $O(\ln T + \min \\{\sigma_{\max}\ln T, \sqrt{d_{tot}} \\})$.

In addition, I suggest modifying the assumption to $d_t \geq 1$, where the feedback for round $t$ is received at round $t+d_t- 1$. This adjustment aligns with the assumptions made in previous works. [1][2]

[1]Quanrud, K. and Khashabi, D. Online learning with adversarial delays. Advances in neural information processing systems, 28, 2015.

[2]Wan, Y., Tu, W.-W., and Zhang, L. Online strongly convex optimization with unknown delays. Machine Learning, 111(3):871–893, 2022.

---

> ### Author Rebuttal · Authors · 2025-04-01
>
> **Q:** Definition of delays and presentation of results.
>
> **A:** In our introduction, we mainly focus on the delay-dependent terms in the bounds for conciseness, but we will clarify the presentation to avoid any confusion. While we also appreciate the suggestion on the definition of delays, we remark that our current definition is common in the related literature. See the seminal work by Weinberger and Ordentlich (2002), Joulani et al. (2013, 2016), and McMahan and Streeter (2014), and more recent work like Masoudian et al. (2022, 2024) and Van der Hoeven et al. (2023).
>
> **Q:** Limited contribution for strongly convex functions and OLR.
>
> **A:** For the *strongly convex* case, we remark that our analysis is **radically different** from the one by Wu et al. (2024), which is crucial in proving the improved guarantees and we believe is an important contribution. As mentioned in our Appendix F and Section 2 as well as in our response to Reviewers jw1R and htSw, the bound we prove via our analysis is essentially optimal, while the results by Wu et al. (2024) have multiple disadvantages; see our responses to Reviewers jw1R and htSw for more details due to the space limit.
> Moreover, our careful regret analysis also leads to our $\sigma_{\max}\ln T$ regret bound, which can be significantly better as $\sigma_{\max}$ can be much smaller than $d_{\max}$ as shown in Lemma B.10.
>
> For the *OLR* setting, we also argue that our contribution is not limited as the analysis required significant changes. In OLR, we consider the unconstrained $\mathcal{X} = \mathbb{R}^n$ and need to leverage the structure to tackle the problem. The main change is in analyzing the $\text{Drift}_T$ term in the regret. While we can nicely control it in the exp-concave case, also due to the bounded diameter, in OLR we cannot do the same. This challenging task requires clipping the predictions and we also avoid prior knowledge of $\max_t |y_t|$, which needs a careful analysis of the cumulative clipping error (see lines 1277-1308 and 1344-1358).
>
> **Q:** Parameter setting and its impact in experiments.
>
> **A:** The learning rates of all baseline methods and our algorithms are proportional to the theoretical ones w.r.t. $T$ and delay-related terms, whereas we ignored $G$ and $D$ as they are reasonable constants in our experimental setup. Following your suggestion, we rerun the experiments with the learning rates as stated in the theoretical results. What we observe is that the performance of all algorithms worsens as including the worst-case bound $G$ and $D$ is somewhat pessimistic, while leading to no change from a comparative point of view, hence providing essentially no further information than the current plots.
>
> **Q:** Validation on real-world data.
>
> **A:** We first remark that the contribution of our work is mainly theoretical, as written in the Impact Statement and observed by Reviewer jw1R. The experiments' goal is thus to validate our findings, which we think already transpires from our current empirical results. Following your suggestion, we consider a real-world dataset [mg\_scale](https://www.csie.ntu.edu.tw/~cjlin/libsvmtools/datasets/regression.html\#mg) with 1385 samples; each sample has 6 features with values in $[-1,1]$ and a label in $[0,2]$. The experimental setup, including constructions of losses and delays, follows what already done for the experiments in our work. The resulting plots are found in the "Real World Data" folder in [this repository](https://anonymous.4open.science/r/Supplementary-experiments-EF2D), and show a similar behavior of the algorithms as already shown in our original experiments.
>
> **Q:** Assumption $t+d_t \leq T$.
>
> **A:** To be precise, we do not make explicit use of it, and we may even remove it. In any case, we remark that one can always introduce it **without loss of generality** because the feedback of any round $t$ with $t+d_t\ge T$ is not used by any learner. This assumption is also made, e.g., by the same authors of [1; Joulani et al., 2013] in Joulani et al. (2016).
>
> **Q:** Improvement w.r.t. the diameter dependence.
>
> **A:** To the best of our knowledge, previous methods (e.g., Wan et al., 2022; Wu et al., 2024) have a polynomial dependence on the diameter $D$ of $\mathcal{X}$, while Wu et al. (2024) also show a $O(\frac{G^2}{\lambda}(d_{\max}+1)\ln T)$ regret via an FTL-based algorithm but only for the significantly *easier* setting with full-function feedback. We remark that the regret of our delayed FTRL as well as our novel analysis for delayed OMD are the first results that are independent of $D$, thus allowing to handle even *unbounded* domains, and recover the optimal $O(\frac{G^2}{\lambda}\ln T)$ regret in the no-delay setting under strong convexity.
>
> **Q:** Typo at line 929.
>
> **A:** Thanks, we will fix it in our next revision.

---

> > ### Comment · Reviewer_zZM9 · 2025-04-04
> >
> > Thanks for the rebuttal. The authors have addressed my concerns; therefore, I decide to increase my score.

---

### Official Review · Reviewer_jw1R · 2025-03-13

**Overall Recommendation:** 3

**Summary:**

In this paper, the authors consider online convex optimization with delayed feedback, and aim to exploit the curvature property of loss functions, i.e., strong convexity and exp-concavity, to improve the regret bound. Specifically, for strongly convex functions, the authors show that a delayed variant of follow-the-regularized-leader can obtain a regret bound of $O(\log T+\min(\sigma_{\max}\log T,\sqrt{d_{tot}}))$. When functions are exp-concave, the authors propose a delayed variant of online Newton step, and establish a regret bound  $O(n\log T+\min(d_{\max}n\log T,\sqrt{d_{tot}}))$. Moreover, the authors also consider the problem of online linear regression, and propose a delayed variant of the Vovk-Azoury-Warmuth forecaster.

---Post Rebuttal---

Thanks for the authors' responses. I agree that the $O(\sigma_{\max}\log T)$ regret bound cannot be *directly* derived by Theorem 2 and Lemma 5 in [1]. However, I still feel this result is not surprising. Thus, I keep my original score.

**Claims And Evidence:**

Yes, all theoretical results have been proved by detailed analysis.

**Essential References Not Discussed:**

Although the authors have cited some existing algorithms for online convex optimization (OCO) with delays and strongly convex functions (Wan et al., 2022; Wu et al., 2024), there exists a delayed variant of online Frank-Wofle (OFW) for strongly convex functions [1][2] that is related to the delayed variant of follow-the-regularized-leader (FTRL) proposed in this paper.

Specifically, OFW originally can be viewed as a combination of follow-the-regularized-leader (FTRL) with linear optimization, and if the projection is allowed, it actually can recover the original FTRL. Therefore, the delayed OFW (i.e., Algorithm 2 in [1]) can also recover the delayed FTRL. For example, it is easy to modify the proof of Theorem 2 and Lemma 5 in [1] to derive a regret bound of $\sigma_{\max}\ln T$ for strongly convex functions.

[1] Wan et al. Projection-free Online Learning with Arbitrary Delays. In arXiv:2204.04964v2, 2023.

[2] Wan et al. Online Frank-Wolfe with Arbitrary Delays. In NeurIPS, 2022.

**Experimental Designs Or Analyses:**

As a theoretical work, the experimental setting is acceptable. A minor concern is that the total number of rounds used in the experiments is too small, i.e., $T=1000$.

**Methods And Evaluation Criteria:**

Yes.

**Other Comments Or Suggestions:**

#Suggestions
1) In Lemma D.1 and its proof, $N$ should be $T$.
2) In line 850, the superscript of $d_{\max}$ is omitted.
3) In line 888, $g_t$ should be $g_{\tau}$

**Other Strengths And Weaknesses:**

#Strengths
1) The authors provide a careful analysis, especially the simple yet useful relaxation of $|m_t|/(t-1)$, to derive an improved regret bound for delayed online convex optimization with strongly convex functions.
2) The authors propose a delayed variant of online Newton step without using the black-box reduction, and establish an $O(n\log T+\min(d_{\max}n\log T,\sqrt{d_{tot}}))$ regret bound for exp-concave functions. The analysis is more complicated than that of strongly convex functions.
3) The authors also consider the problem of online linear regression, and develop a delayed variant of the Vovk-Azoury-Warmuth forecaster.

#Weaknesses
1) There lacks a detailed comparison with the theoretical results derived by the black-box method (Joulani et al., 2013). Moreover, it seems that the black-box method can achieve $O(\sigma_{\max}n\ln T)$ regret bound for exp-concave functions. However, the regret bound of this paper depends on the maximum delay $d_{\max}$, instead of $\sigma_{\max}$.
2) As also confirmed by the authors, the existing algorithm of Wu et al. (2024) is sufficient to enjoy the improved regret bound for strongly convex functions, which limits the significance of the delayed variant of follow-the-regularized-leader (FTRL). Moreover, there exists a delayed variant of online Frank-Wofle that is related to the delayed FTRL but missed.
3) Some procedures in the proofs are unnecessary. For example, below Eq. (17), from the definition of $x_t^\ast$, it is easy to verify that $(F_t^\ast(x_t^\ast)+\langle g_t, x_t^\ast \rangle)-(F_t^\ast(x_{t+1}^\ast)+\langle g_t, x_{t+1}^\ast \rangle)\leq \langle g_t,x_t^\ast- x_{t+1}^\ast \rangle$ (it is no need to use the convexity of $F_t^\prime$, the definition of $F_t^\prime$, and, the first-order optimality). The same issue exists below Eq. (26). Moreover, some procedures need more explanations, e.g., the straightforward calculations for Eq. (17).

**Questions For Authors:**

At the bottom of page 29, if we simply combine Eq. (56) and (58), it seems that $\eta_t^2$ will be introduced, which is different from the procedures in your Eq. (59). Is there a typo somewhere？

**Relation To Broader Scientific Literature:**

This paper is related to existing works about online convex optimization (OCO) with curvature and/or delayed feedback. Among those related works, the most comparable ones are existing results for OCO with strongly convex functions and unknown delays (Wan et al., 2022; Wu et al., 2024) and a black-box method for OCO with arbitrary but stamped delays (Joulani et al., 2013). Although the authors have cited these works, the corresponding discussions are a bit opportunistic.

Specifically, one actually can simply utilize the black-box method (Joulani et al., 2013) to derive an $O(\sigma_{\max}n\ln T)$ regret bound for delayed OCO with exp-concave functions respectively, and a similar result for online linear regression. However, in Table 1 (as well as the whole Introduction), it seems that there are no existing results on exp-concave functions and online linear regression. I understand that the authors may want to emphasize the black-box method is not suitable for the case with unknown delays, i.e., the timestamp of each feedback is unknown. But, their regret bound for exp-concave functions also needs to know the timestamp of each feedback.

In the case of strongly convex functions, the authors emphasize that their result has two improvements, i.e., replacing $d_\max$ in the regret bound with $\sigma_\max$ and simultaneously achieving a regret bound of $\ln T+\sqrt{d_{tot}}$. However, as also confirmed by the authors, the existing algorithm of Wu et al. (2024) is sufficient to enjoy the improved regret bound, which limits the significance of the proposed algorithm for strongly convex functions. Moreover, from the technical view, it is actually trivial to replace $d_\max$  with  $\sigma_\max$ (based on an existing work discussed below). The only interesting finding is that the term $|m_t|/(t-1)$ can be relax to $|m_t|/(\sum_{\tau\leq t}|m_{\tau}|)$, and then the sum of this term over $t=1,...,T$ can be bounded by $\sqrt{d_{tot}}$ based on a classical inequality.

**Theoretical Claims:**

Yes, I have checked almost all the proofs in this paper, and do not find any serious problems.

---

> ### Author Rebuttal · Authors · 2025-04-01
>
> **Q:** $T=1000$ is too small in experiments.
>
> **A:** We extended our experiments to have $T=10000$ and plan to include the new plots in our next revision. The new plots essentially show the same behavior with an extended time horizon. You can find these extra plots in the "Synthetic Data" folder at [this repository](https://anonymous.4open.science/r/Supplementary-experiments-EF2D).
>
> **Q:** Comparison to the black-box method (Joulani et al., 2013).
>
> **A:** We thank the reviewer for pointing this out. Indeed, applying the black-box reduction (Joulani et al., 2013) with ONS can achieve $O(\sigma_{\max}n\ln T)$ regret, and our Algorithm 2 also requires the timestamps to compute the regularizer. However, in Table 1 we did not compare with the algorithms based on the black-box reduction due to their typical inefficiency (also shown to some extent by our experiments). Nevertheless, we will add these remarks for the exp-concave and online linear regression settings in our discussion as well as Table 1 to have a more comprehensive comparison with existing techniques.
>
> **Q:** Improved regret bounds under strong convexity and comparison with Wu et al. (2024).
>
> **A:** In fact, proving that the algorithm proposed by Wu et al. (2024) achieves the optimal regret bound in OCO with delays is one of the contributions of our work. Note that our analysis is **radically different** from Wu et al. (2024), which was crucial in deriving the improved guarantees and we believe is an interesting contribution. As mentioned in our Appendix F and Section 2 as well as in our response to Reviewers zZM9 and htSw, the bound we prove via our analysis is essentially optimal, while the bound $O(\frac{G^2+D}{\lambda}(d_{\max}+1)\ln T + \frac{G}{\lambda^2}(d_{\max}+1))$ in Wu et al. (2024) has some disadvantages including: (1) a multiplicative dependence on the diameter $D$ of $\mathcal{X}$; (2) no robustness to the $\sqrt{d_{tot}}$ regime; (3) a worse dependence on the strong convexity parameter $\lambda$; (4) a worse $d_{\max}$ dependence ($d$ in their work is $d_{\max}+1$ in our notation). The improvements of our analysis are thus multiple, with no assumption on the domain $\mathcal{X}$ having a bounded diameter $D$, thus proving that the algorithm can work even in **unbounded** domains, the optimal dependence on $G$ and $\lambda$ hence recovering the optimal $O(\frac{G^2}{\lambda}\ln T)$ regret bound in the no-delay setting, and the improved robustness to delays given by our key observations on $\sum_t |m_t|/(t-1)$.
>
> **Q:** Improvement from $d_{\max}$ to $\sigma_{\max}$ is trivial based on OFW in [1].
>
> **A:** We respectfully disagree with the reviewer that the improvement from $d_{\max}$ to $\sigma_{\max}$ is trivial. While this refinement may appear straightforward in hindsight, to the best of our knowledge, no prior work has provably achieved a $\sigma_{\max}\log T$ regret for strongly convex losses. Moreover, the improvement is significant as $\sigma_{\max}$ can be much smaller than $d_{\max}$ as shown in Lemma B.10. We thus believe our improvement is valuable. As for the extra references on projection-free algorithms for OCO with delays, we thank the reviewer for providing them and will incorporate them in our next revision. However, we wonder whether this is directly related to the above improvement as Theorem 2 in [1] only achieves $O(\frac{G^3+\lambda^3D^3}{\lambda}(d_{\max}+1)\ln T+\frac{G^2+\lambda^2D^2}{\lambda}T^{2/3})$ regret ($d$ in [1] is $d_{\max}+1$ in our notation), which does not involve $\sigma_{\max}$ and has a polynomial dependence on the diameter $D$ of the action set as well as a worse dependence on $G$, let alone the clearly suboptimal $T^{2/3}$ term. As for Lemma 5 in [1], we are also not sure whether this is directly related since we consider a single update using all the gradients received as feedback in each round and do not bound terms similar to $\\|y_{\tau_{c_t}}-y_t\\|_2$; Lemma 5 also seems to present some downsides that end up causing the above-mentioned shortcomings of Theorem 2, which seem unavoidable via the projection-free algorithms in [1]. We would appreciate it if the reviewer could kindly clarify further on this comment.
>
> **Q:** Explanations to Eq. (17).
>
> **A:** We solve the quadratic inequality w.r.t. $\\|x_t^*-x_{t+1}^*\\|_2$ from the previous math display and relax its upper bound to derive (17). We will make this clearer.
>
> **Q:** Eq. (56)-(59) and dependence on $\eta_t$.
>
> **A:** Thanks for pointing out the typo. The first inequality in (56) is missing $\eta_t$ in the r.h.s.
>
> **Q:** In Lemma D.1, $N$ should be $T$.
>
> **A:** The lemma holds for any $N \in [T]$ and we use it with $N = \tau^\star < T$ in the proof of Corollary 4.2. We will quantify $N$ in the statement of Lemma D.1 and clarify its usage.
>
> **Q:** Other comments and typos.
>
> **A:** Thanks for pointing out other minor typos and simplifying the derivation below Eq.(17)/(26). We will incorporate these fixes in our next revision.

---

### Official Review · Reviewer_CNFg · 2025-03-14

**Overall Recommendation:** 4

**Summary:**

The authors present a FTRL-based algorithm that achieves logarithmic regret for strongly convex loss functions. More importantly it depends on $\min(\sigma_{max} \log T, \sqrt{d_{tot}})$ which improves the previous results $O(\sqrt{d_{max}} \log T)$. The key idea is to have a regularizer that uses all the previous timesteps and not just the ones with observations. This helps bound the difference between the update $x_t$ with delays and the hypothetical $x_t^\star$ with no delays.

Furthermore they present an algorithm, inspired from ONS, for exp concave losses, which is the first algorithm to give regret when there is delay in the observation of the gradients. They achieve $\min(d_{max} n \log T, \sqrt{d_{tot}})$.

Finally, they present an adaptation of Vovk-Azoury-Warwuth for online linear regression, which is also the first algorithm to solve it with delay.

## update after rebuttal

Based on the successful adversarial  experiences, I decided to maintain my score.

**Claims And Evidence:**

They provide proofs for all their claims and show with an experiment, how their algorithm performance does depend on that minimum: $\min(d_{max} n \log T, \sqrt{d_{tot}})$.

**Essential References Not Discussed:**

None that I can think of.

**Experimental Designs Or Analyses:**

The experiment setting is a good one, except for the data generation that is NOT adversarial but stochastic with a fixed distribution. It would be way more compelling to use a distribution that changes, potentially in a way to disturb the proposed method as much as possible.

**Methods And Evaluation Criteria:**

They use regret as a metric which is standard. Their algorithms also follow standard existing algorithms.

**Other Comments Or Suggestions:**

Typo
- line 628: $|m_{t'}| \to |m_{t^\star}|$
- line 850: $d_{max} \to d_{max}^{\leq N}$

**Other Strengths And Weaknesses:**

Strengths:
1- Provides the first algorithm for  exp-concave and OLR cases
2- Strictly improves  previous methods for strongly convex by having a bound that depend on the minimum of two quantities that, depending on the setting, can differ significantly
3- The paper is clear and quite easy to follow

Weaknesses:
1- The experiment is fully stochastic and not adversarial enough.

**Questions For Authors:**

- can you run the experiments on more adversarial settings?
- minor question: On line 583, why use $A_1 + A_1^T$ for the Hessian instead of $2A_1$ since the matrix $A_1$ is symmetric?

**Relation To Broader Scientific Literature:**

The algorithms build on existing ones for standard OCO without delay, i.e FTRL, ONS and VAW.

They then modify the regularizers and the learning rate to adapt for the delay in feedback, which is new.

Their algorithm for strongly convex has some similarities with Wan et al. (2022) and Wu et al. (2024) with the major difference in that they update $x_t$ and every timestep (even in the absence of new gradients) and use a regularizer that cover all previous timesteps, not just the ones already observed.

**Theoretical Claims:**

I checked the proofs of the strongly convex and exp-concave claims. I did not check the proof of the OLR problem. I did not see anything problematic in the first two.

---

> ### Author Rebuttal · Authors · 2025-04-01
>
> **Q:** Can you run the experiments on more adversarial settings?
>
> **A:** Following your suggestion, we run our benchmark algorithms on the following more adversarial (i.e., non-stationary) environment. Specifically, in this environment, we set the time horizon as $T = 10000$ (as to also address a concern by Reviewer jw1R) and we partitioned rounds into roughly $\log_2 T$ phases where the length of phase $s \ge 1$ is $2^{s-1}$. For $t\in\\{2^{s-1},\dots,2^s-1\\}$, when $s$ is odd, the feature vector $z_t$ is sampled from the multivariate standard Gaussian distribution with values clipped to the range $[-1,1]$ and the delay $d_t$ is independently sampled from a geometric distribution with a success probability of $T^{-1/3}$; when $s$ is even, each coordinate of $z_t$ is independently and uniformly sampled from $[-1,1]$ and the delay $d_t$ is independently and uniformly sampled from the set $\\{0,1,\dots,5\\}$. The loss function is designed in the same way as already described in Section 6 of our paper for each of the three settings, except we additionally clip the standard Gaussian noise to $[-1,1]$ to have a bounded gradient. You can find the additional plots for these experiments in the "Adversarial Data" folder at the following anonymized link [https://anonymous.4open.science/r/Supplementary-experiments-EF2D](https://anonymous.4open.science/r/Supplementary-experiments-EF2D). We will integrate these plots with appropriate comments, as well as a reference to the source code, in the next revision of our paper.
>
> **Q:** Typos at lines 583, 628, and 850.
>
> **A:** Thanks for carefully pointing out these typos. We will fix them in our next revision.

---

> > ### Comment · Reviewer_CNFg · 2025-04-04
> >
> > My commet on the adversary was less on the delay and more on the sampling of your data. They are basically gaussian everywhere, meaning an algorithm that minimizes for that would perform as well. I would expect you to challenge a bit more by modifying the data so that there is a big shift in the minimum between phases or even within phase, by having your data drifting for example. This, in my option, remains more of a stochastic setting than an adversarial one.

---

> > > ### Author Response · Authors · 2025-04-06
> > >
> > > We thank the reviewer for the further response. Based on your suggestion, we designed a new non-stationary environment as follows. The generation processes for feature vectors and delays, as well as the definition of the loss function, remain the same as the previous environment discussed in the rebuttal. However, we modified the generation of the label $y_t$:
> > > $$y_t=\bigl\langle z_t,\theta_t \bigr\rangle+\epsilon_t\,$$
> > > where the latent vector $\theta_t$ alternates every 30 rounds between the two vectors $\mathbf{1}$ and $\mathbf{0}$. This periodic change introduces non-stationarity, reflecting scenarios where the optimal action shifts over time.
> > > Additionally, we also modify the noise term
> > > $\epsilon_t$ inspired by Xu and Zeevi (2023). Specifically, we flatten an abstract art piece by Jackson Pollock and take consecutive grayscale values in $[0,1]$ as the noise $\epsilon_t$. If you look at this artwork, you will see that the noise $\epsilon_t$ is inherently adversarial. In this environment, the optimal action may change every 30 time steps.
> > > You can find both the artwork and the additional plots for these experiments (with both $T=1000$ and $T=10000$) at the following anonymous link [https://anonymous.4open.science/r/Supplementary-experiments-adv-13DC/](https://anonymous.4open.science/r/Supplementary-experiments-adv-13DC/). The results show that our algorithms still perform the best among all the benchmark algorithms. We will incorporate these plots in the next revision of our paper.
> > >
> > > **References:**
> > > - Xu, Y. and Zeevi, A. (2023): "Bayesian design principles for frequentist sequential learning", ICML 2023.

---

### Decision · Program_Chairs · 2025-05-01

**Decision:**

Accept (poster)

**Comment:**

This paper provides an algorithm for online convex optimization with delayed feedback in the setting of exp-concave losses and online linear regression. The regret bounds appear to improve the delay-dependent terms of previous work from a worst-case dependence on the maximum delay, to a dependence on the sum of the delays. Overall, the reviews express some concern the closeness of this paper's contribution to prior work. The authors defend this point by pointing out deficiencies in prior analyses, and all reviewers are in favor of acceptance.
The authors are encouraged to incorporate their reviewer response into the final copy.